# Small protein modules dictate prophage fates during polylysogeny

Justin E. Silpe[1,2], Olivia P. Duddy[1], Grace E. Johnson[1,2], Grace A. Beggs[1], Fatima A. Hussain[3], Kevin J. Forsberg[4] & Bonnie L. Bassler[1,2 ✉]

Most bacteria in the biosphere are predicted to be polylysogens harbouring multiple prophages[1–5]. In studied systems, prophage induction from lysogeny to lysis is near-universally driven by DNA-damaging agents[6]. Thus, how co-residing prophages compete for cell resources if they respond to an identical trigger is unknown. Here we discover regulatory modules that control prophage induction independently of the DNA-damage cue. The modules bear little resemblance at the sequence level but share a regulatory logic by having a transcription factor that activates the expression of a neighbouring gene that encodes a small protein. The small protein inactivates the master repressor of lysis, which leads to induction. Polylysogens that harbour two prophages exposed to DNA damage release mixed populations of phages. Single-cell analyses reveal that this blend is a consequence of discrete subsets of cells producing one, the other or both phages. By contrast, induction through the DNA-damage-independent module results in cells producing only the phage sensitive to that specific cue. Thus, in the polylysogens tested, the stimulus used to induce lysis determines phage productivity. Considering the lack of potent DNA-damaging agents in natural habitats, additional phage-encoded sensory pathways to lysis likely have fundamental roles in phage–host biology and inter-prophage competition.

Phages are viruses that infect bacteria and are important drivers of bacterial diversity and microbial community development[7,8]. Following host infection, temperate phages can undertake one of two lifestyles. They can enter the lytic cycle, whereby they exploit host resources, replicate, produce viral particles and kill their host[6,9]. Alternatively, phages can enter the lysogenic state and remain dormant (that is, as prophages) and be passed down to host progeny[10]. Phages can also cause chronic host infections in which they persist and extrude viral particles without killing the host[11,12].

During lysogeny, model temperate phages, including phage lambda, produce a repressor (called cI) that binds and prevents expression from a promoter (termed $P_R$) that controls the lysis genes[9]. Following DNA damage, activation of the host RecA protein leads to auto-proteolysis and inactivation of the cI repressor[13,14]. Consequently, $P_R$ is de-repressed, which triggers phage replication, host-cell lysis and transmission of the phage to neighbouring bacteria. Other temperate phages harbour repressors that lack the peptidase domain responsible for autoproteolysis. Instead, a peptidase or antirepressor encoded elsewhere in the phage genome is activated by the host SOS response[15,16]. The understanding that all bacteria possess *recA*, coupled with the fact that phages are omnipresent, has led to the common view that the host SOS response is the universal prophage inducer. However, significant concentrations of potent DNA-damaging agents are rare in the environment and, increasingly, phages are being discovered that are not induced by DNA damage[17]. Together, these findings suggest that undiscovered induction triggers exist in nature.

Recent findings have revealed that quorum-sensing (QS) signals represent one SOS-independent induction trigger for phage lysis–lysogeny lifestyle transitions[18–20]. QS is a process of cell-to-cell communication that bacteria use to orchestrate collective behaviours. QS relies on the production, release and group-wide detection of and response to extracellular signalling molecules called autoinducers (AIs)[21]. Phages can harbour phage-to-phage QS-like communication systems, such as the arbitrium system in SPβ phages[20]. Alternatively, as in vibriophage VP882, they can monitor host bacterial QS-mediated communication pathways to tune the timing of the lysogeny-to-lysis switch to changes in host-cell density[19]. Phage VP882 is a linear plasmid-like prophage that encodes a homologue of the *Vibrio* QS receptor VqmA (called VqmA_Phage), which is activated by a host-produced AI called DPO[19,22]. Following binding to DPO, VqmA_Phage activates the transcription of a counter-oriented gene called *qtip*. Production of Qtip induces host-cell lysis. Our hypothesis is that through the surveillance of a bacterial-produced QS signal, the phage can integrate host-cell density information into its decision-making process. Moreover, by lysing its host at high cell density, the phage maximizes the probability of infecting other cells in the population.

Bacteria commonly harbour multiple prophages, a state called polylysogeny. How prophages residing in a polylysogenic host compete for host resources is not well understood. What is known is that following DNA damage, the number of virions released for lambdoid prophages is lower in polylysogens than in monolysogens[23]. This result suggests that in polylysogens, co-induced prophages compete for reproductive

[1]Department of Molecular Biology, Princeton University, Princeton, NJ, USA. [2]Howard Hughes Medical Institute, Chevy Chase, MD, USA. [3]Department of Civil and Environmental Engineering, Massachusetts Institute of Technology, Cambridge, MA, USA. [4]Department of Microbiology, University of Texas Southwestern Medical Center, Dallas, TX, USA. ✉e-mail: bbassler@princeton.edu

success. It is possible that in the context of polylysogeny, if a prophage possesses an alternative, non-DNA damage-dependent pathway to lysis, it could compete more effectively with co-residing prophages that cannot respond to the alternative induction cue. Here we sought to identify and characterize polylysogenic bacteria that harbour prophages that possess SOS-dependent and SOS-independent pathways to lysis. Our aim was to use them as models to explore within-host prophage competition.

The current work describes the discovery of SOS-independent, phage-encoded lysis-inducing modules that, despite bearing little resemblance to one another at the sequence level, share a common regulatory logic. Our characterization of several of these phages shows that they all use a transcription factor (TF) to activate the expression of a divergently transcribed gene encoding a small protein (smORF). The smORF induces the transition from lysogeny to lysis. The smORFs studied here lack homologues and predictable domains; however, they operate by inactivating the same respective target, the cI repressor, in the phage that encodes the smORF. The mechanisms by which the TFs regulate their partner *smORF* genes vary. In some cases, the TFs operate independently. In other cases, *smORF* expression requires a xenobiotic responsive element (XRE) family protein working in conjunction with a LuxR-type QS receptor or TF that requires a bacterial-produced AI ligand for activity.

The prophage-containing isolates on which we focus are polylysogenic. We show that the addition of a DNA-damaging agent leads to phage-mediated lysis of the bacteria and the release of a mixed population of phage particles. Single-cell analyses demonstrate that this outcome stems from a subset of host cells expressing lytic genes from only one of the phages, another subset expressing lytic genes from only the other phage and the final subset of cells expressing lytic genes from both phages. Unlike DNA damage, induction through the newly discovered regulatory modules results in gene expression from and near-exclusive production of the phage responsive to the specific input. Our results suggest that the activities of these SOS-independent pathways dictate the outcomes of prophage–prophage competition by expanding the range of stimuli to which specific prophages can respond.

## Two mechanisms of phage-driven lysis

To advance studies of inter-prophage competition in polylysogenic bacteria, we conducted a search among sequenced phage genomes for genes encoding putative SOS-independent lysis–lysogeny modules located between *repA* and *telN*, which are hallmarks of all known linear plasmid-like phages. Our strategy was inspired by the arrangement of the *vqmA*$_{Phage}$ and *qtip* genes, which are encoded between *repA* and *telN* in vibriophage VP882 (ref. 19). We searched all prophage genomes at the National Center for Biotechnology Information (NCBI) database and six recently curated phage and phage-plasmid databases[24–29]. The genomes span diverse environmental, marine and human body sites, and we searched for convergently oriented *repA* and *telN* genes residing within 10 kb of each other. The search revealed 784 putative linear plasmid-like phage genomes, 274 of which contained distinct yet conserved *repA–telN* loci (Supplementary Table 1). In 271 out of the 274 genomes (99%), the gene upstream of *repA* encoded a cI-like DNA-binding protein adjacent to putative, divergently transcribed lytic, structural and regulatory genes. A panel of representative loci is shown in Fig. 1a. We constrained our search to phage genomes that harboured RecA-dependent, autoproteolytic cI repressors because autoproteolytic repressors exhibit a stereotypical response to SOS (that is, repressor cleavage), which is bioinformatically predictable and testable in vitro. Non-proteolytic cI repressors require additional phage-specific and/or host-specific factors for regulation. Therefore, we did not consider them for the current study. For phages that possess autoproteolytic cI repressors, we reasoned that any additional regulatory modules uncovered in their

genomes would likely respond to SOS-independent inputs. Filtering phage genomes using these criteria led to 61 distinct loci. The majority of phages eliminated at this step (210 out of 271) encoded *repA–telN* loci that resembled those of the *Escherichia coli* linear plasmid-like phage called N15. This finding is likely a consequence of the overrepresentation of *Enterobacteriaceae* in the sequencing databases. N15 is known to be subject to antirepression[30,31], and we determined that the repressor it encodes is non-proteolytic (Fig. 1b). The only member within the identified set of phages that harboured autoproteolytic repressors that had been previously investigated was phage VP882 (Fig. 1a,b), described above, and a singleton in our current cluster analysis (Supplementary Table 1). Many of the phages from our database searches come from metagenomics sequencing projects, are unobtainable and/or have no known host. Nonetheless, with the goal of probing the functions of these putative sensory input pathways, we focused on obtainable phage–host pairs.

## TF–smORF modules are SOS independent

The first isolate we investigated was *Vibrio cyclitrophicus* 1F-97 (henceforth called *Vibrio* 1F-97), which encodes a putative phage on genome contig 72 (hereafter called phage 72). Between the phage 72 *repA* and *telN* genes are genes encoding a putative TF (as judged by its predicted DNA-binding domain according to the NCBI conserved domain database) and a counter-oriented small, 171 nucleotide open reading frame (ORF) (hereafter called TF$_{72}$ and smORF$_{72}$, respectively). The sequences of these genes lacked nucleotide-level and amino acid-level identity to the *vqmA*$_{Phage}$ and *qtip* genes. However, their arrangement paralleled that of *vqmA*$_{Phage}$ and *qtip* in phage VP882. To test whether the DNA located between *repA* and *telN* on phage 72 controls the phage lysis–lysogeny transition, *smORF*$_{72}$ was cloned under the control of an anhydrotetracycline (aTc)-inducible promoter on a plasmid (pTet–*smORF*$_{72}$) and conjugated into *Vibrio* 1F-97. Addition of aTc to this recombinant strain led to a precipitous decline in the optical density at 600 nm (OD$_{600}$), similar to when the potent SOS-activator ciprofloxacin was added. This result indicated that phage-driven lysis occurred (Fig. 1c). aTc administered to *Vibrio* 1F-97 harbouring an empty vector, or to *Vibrio cholerae*, *Vibrio parahaemolyticus* or *E. coli* (none of which contain phage 72) harbouring the pTet–s*mORF*$_{72}$ vector, did not affect growth. This result showed that the smORF$_{72}$ protein alone does not induce lysis (Extended Data Fig. 1a,b). We eliminated the possibility that an additional predicted 165 nucleotide ORF encoded between *repA* and *telN* functions similarly to smORF$_{72}$, as overexpression of the gene encoding this protein did not drive lysis (Fig. 1a and Extended Data Fig. 1c). Thus, host-cell lysis requires the presence of the phage and induction of the *smORF*$_{72}$ gene. Supplementation with ciprofloxacin did not activate *smORF*$_{72}$ expression in wild-type *Vibrio* 1F-97 (Extended Data Fig. 1d), which suggests that smORF$_{72}$ production is independent of SOS. Notably, phage preparations obtained from *Vibrio* 1F-97 treated with aTc or with ciprofloxacin contained phage 72 particles, which indicates that phage 72 can be induced by both SOS-independent and SOS-dependent pathways (Extended Data Fig. 1e).

We next explored how induction of smORF$_{72}$ promotes lysis. Important for this step is that our bioinformatics analysis revealed that 99% of the *repA–telN* loci (271 out of 274) contained genes upstream of *repA* encoding predicted phage-repressor proteins (called cI$_{72}$ and its target promoter P$_{R72}$, respectively, for phage 72). We verified the function of this repressor–promoter pair by fusing the P$_{R72}$ promoter to *lux* on a plasmid (P$_{R72}$–*lux*). Recombinant *E. coli* carrying the construct made light, which decreased by 500-fold when the gene encoding cI$_{72}$ was also introduced (Extended Data Fig. 1f). Consistent with this result, purified cI$_{72}$ protein bound to the P$_{R72}$ promoter in vitro (Extended Data Fig. 1g). Introduction of pTet–*smORF*$_{72}$ into *E. coli* carrying the *cI*$_{72}$–P$_{R72}$–*lux* plasmid restored high level light production, which indicated that production of smORF$_{72}$ inactivates the cI$_{72}$ repressor (Fig. 1d). Unlike DNA

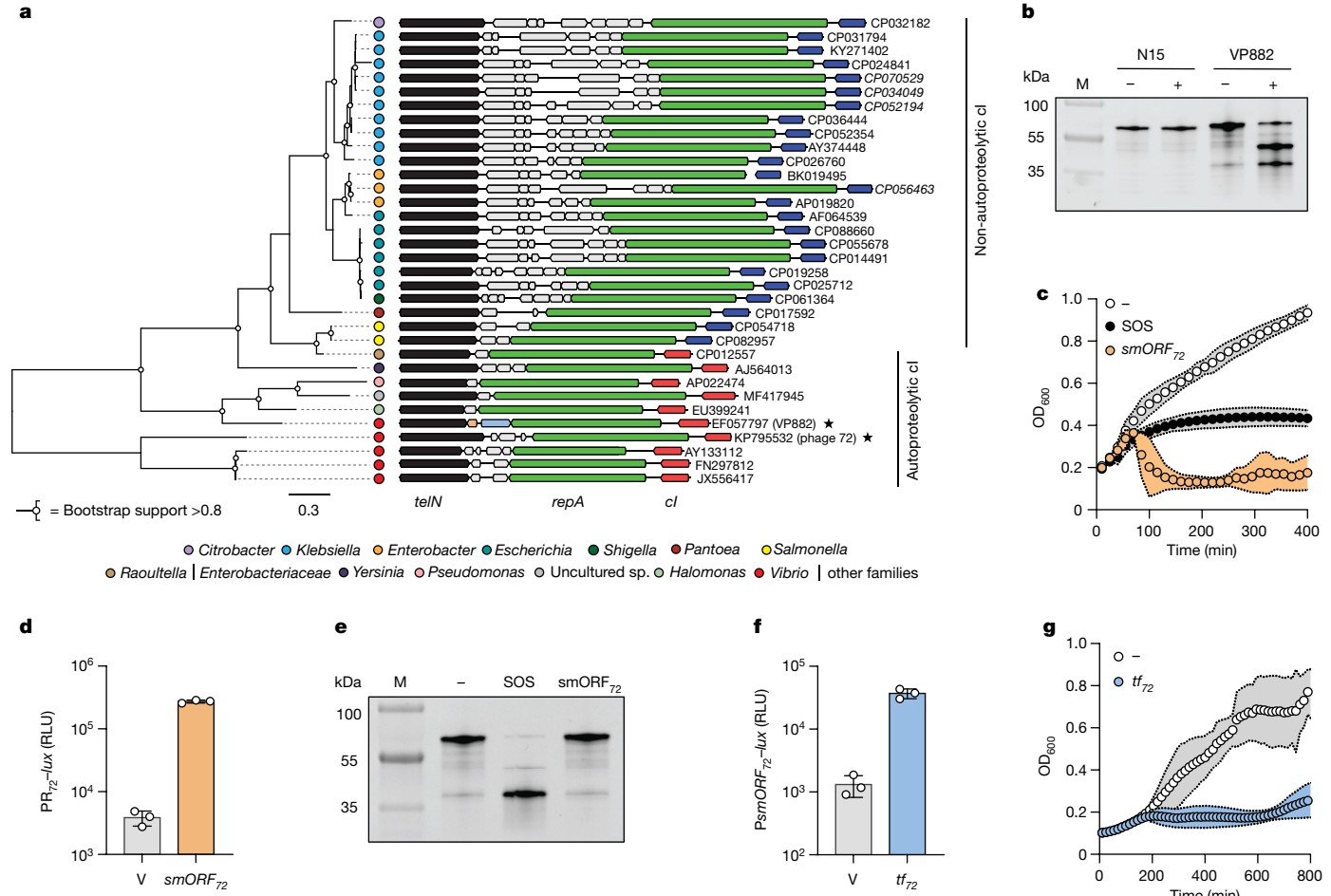

**Fig. 1 | Variable gene content in an otherwise conserved locus of linear plasmid-like phages reveals TF–smORF modules that regulate lysis independently of SOS. a**, Phylogenetic tree (left) and gene neighbourhoods (right) for 34 representative *telN* loci encoding convergently oriented *telN* and *repA* genes. Phage VP882 and *Vibrio* 1F-97 phage 72 are denoted with stars. The SOS-independent pathway components in phage VP882 (*vqmA_Phage* and *qtip*) are in blue and orange, respectively. Genes encoding predicted autoproteolytic repressors (red) cluster together with respect to TelN phylogeny and are distinct from genes encoding non-cleavable repressors (navy). NCBI accession numbers are depicted to the right of each sequence. Scale bar indicates the number of amino acid substitutions per site. **b**, SDS–PAGE in-gel labelling of the non-proteolytic N15 phage repressor (HALO–cI_N15) and the autoproteolytic phage VP882 repressor (HALO–cI_VP882). The minus and plus symbols indicate the absence and presence, respectively, of 500 ng ml$^{-1}$ ciprofloxacin used to induce the SOS response. M denotes the molecular weight marker (representative bands are labelled). **c**, Growth of *Vibrio* 1F-97 carrying aTc-inducible *smORF_72* in medium containing aTc (denoted *smORF_72*), ciprofloxacin (denoted SOS) or water (denoted by the minus symbol). **d**, P_R72–*lux* expression in *E. coli* carrying an empty vector (designated V) or aTc-inducible *smORF_72* grown in medium containing aTc. The P_R72–*lux* plasmid carries *cI_72*, which represses reporter expression. Relative light units (RLUs) were calculated by dividing bioluminescence by OD_600. **e**, SDS–PAGE in-gel labelling of the *Vibrio* 1F-97 phage 72 repressor (cI_72–HALO) produced by *E. coli* carrying aTc-inducible *smORF_72*. The treatments –, SOS and smORF_72 refer to water, ciprofloxacin and aTc, respectively. M as in **b**. **f**, P*smORF_72*–*lux* expression from *E. coli* carrying an empty vector (V) or aTc-inducible *tf_72* grown in medium containing aTc. RLU as in **d**. **g**, Growth of *Vibrio* 1F-97 carrying aTc-inducible *tf_72* in medium lacking or containing aTc (white and blue, respectively). Data are represented as the mean ± s.d. with *n* = 3 biological replicates (**c**,**d**,**f**,**g**) or as a single representative image chosen from three replicates experiments (**b**,**e**).

damage, smORF_72 production did not lead to cI_72 proteolysis (Fig. 1e). These results suggest that smORF_72 is an antirepressor that inactivates its partner cI_72 repressor protein through a nonproteolytic mechanism.

As noted above, a gene encoding a TF, *tf_72*, lies adjacent to *smORF_72*. TF_72 is a logical candidate to be the regulator of *smORF_72* expression. Indeed, production of TF_72 in *E. coli* harbouring the P*smORF_72*–*lux* plasmid (in which the promoter of *smORF_72* is fused to *lux*) increased light output by 28-fold compared with the empty vector (Fig. 1f). Similarly, production of TF_72 in *Vibrio* 1F-97 led to an increase in *smORF_72* transcription (Extended Data Fig. 1h), after which the culture lysed (Fig. 1g). These data suggest that TF_72, through transcriptional activation of *smORF_72*, drives phage-72-mediated lysis.

A central question is how *tf_72* expression is naturally regulated in phage 72 to induce the smORF_72-mediated lysis cascade. In an effort to identify small-molecule inducers of phage 72, we introduced

P*smORF_72*–*lux* into the *Vibrio* 1F-97 lysogen and we monitored both P*smORF_72*–*lux* activity and growth (through OD_600 measurements) following exposure to various compounds. We tested commercially available compound libraries (Biolog MicroArrays, around 2,000 conditions) and a curated library of antibiotics (a gift from the Seyedsayamdost Group, around 250 conditions). Among the antibiotic library compounds are known phage inducers, including DNA-damaging agents and regulators of reactive oxygen species. A reporter for *E. coli* phage lambda induction (P_Rlambda–*lux*) was likewise assessed, which enabled us to determine whether inducers of P*smORF_72*–*lux* and/or inhibitors of growth were specific for phage 72 or not. No test compound elicited more than a twofold increase in P*smORF_72*–*lux* expression or a decline in OD_600 that was specific to phage 72 (Extended Data Fig. 2a). Thus, our screens did not produce potential inducers. Identifying induction stimuli for prophages that are non-SOS-inducible has proven difficult[17].

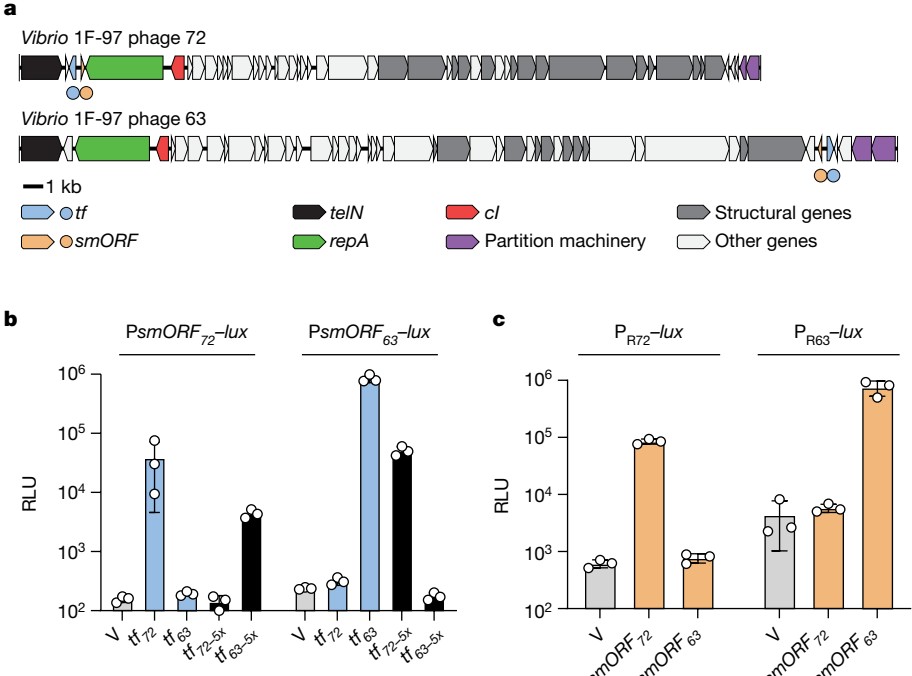

**Fig. 2 | *Vibrio* 1F-97 harbours two linear plasmid-like phages that control lysis through TF–smORF modules encoded in different genomic loci.**
**a**, Organization of genes on linear plasmid-like phages in *Vibrio* 1F-97. Genes are coloured by functional annotation as noted in the key. Circles below genomes denote positions of the *tf* (blue) and *smORF* (orange) genes located at *repA–telN* (phage 72) or by *par* genes (phage 63). All analogous genes, with the exception of those encoding the TFs, share less than 30% amino acid identity (BLASTp).
**b**, P*smORF–lux* output from *E. coli* harbouring P*smORF_{72}–lux* or P*smORF_{63}–lux* and a second plasmid encoding an empty vector (V), aTc-inducible *tf_{72}*, *tf_{63}* or the chimeric *tf* genes, designated *tf_{72-5x}* and *tf_{63-5x}*, each grown in medium

containing aTc. The chimeric TF proteins (black bars) have five amino acids in their DNA-binding domains replaced from the other TF. Thus, TF_{72-5x} possesses five amino acids from TF_{63} and TF_{63-5x} possesses five amino acids from TF_{72}. Extended Data Fig. 3a shows the exchanged residues and their locations.
**c**, Light production from *E. coli* harbouring the cI-repressed reporter plasmids *cI_{72}*-P_{R72}–*lux* (P_{R72}–*lux*) or *cI_{63}*-P_{R63}–*lux* (P_{R63}–*lux*) and a second plasmid encoding an empty vector (V), aTc-inducible *smORF_{72}* or aTc-inducible *smORF_{63}*, each grown in medium containing aTc. Data are presented as the mean ± s.d. with *n* = 3 biological replicates (**b**,**c**). RLU as in Fig. 1d (**b**,**c**).

However, some phenothiazines, including common antipsychotic drugs, which have not previously been implicated in phage induction, drove low-level activation of both phage reporters (Extended Data Fig. 2b).

## *Vibrio* 1F-97 harbours two prophages

Bioinformatics searches (BLASTp) of the phage-72-encoded TF–smORF module indicated that the closest homologue of the TF_{72} regulatory protein in NCBI (about 53% amino acid pairwise identity; Extended Data Fig. 3a) resided on a different contig within the same *Vibrio* 1F-97 genome, contig 63. Contig 63, like phage 72, harboured signatures of a linear plasmid-like phage (for example, genes analogous to *telN* and *repA* by synteny; Fig. 2a). Furthermore, DNA corresponding to contig 63 was present in phage preparations of ciprofloxacin-treated *Vibrio* 1F-97 cultures (Extended Data Fig. 3b). We now refer to this element as phage 63, and we classify *Vibrio* 1F-97 as polylysogenic for phage 63 and phage 72. The two phages shared little identity on a whole-genome basis (26.4% average nucleotide identity), and all analogous genes, with the exception of the two TFs, shared less than 30% amino acid identity (BLASTp). Furthermore, unlike phage 72, cloning and expression of the phage 63 DNA that intervened *repA* and *telN* did not induce lysis (Extended Data Fig. 3c,d). Rather, the gene encoding the homologous TF in phage 63 (*tf_{63}*) was located between genes encoding predicted partition machinery (*parAB*) and an operon encoding predicted late genes (tail and assembly genes; Fig. 2a). *tf_{63}* resided adjacent to, but in the opposite orientation from, a gene encoding another hypothetical 183 nucleotide smORF (*smORF_{63}*). smORF_{63} and smORF_{72} have only 9 identical residues in pairwise alignment (11% amino acid identity;

Extended Data Fig. 3e). For phage 63, we constructed the analogous set of tools described above for phage 72 and determined that the *par*-associated *tf_{63}–smORF_{63}* pair encoded proteins that performed equivalent functions as the *tf_{72}–smORF_{72}* pair (Extended Data Fig. 4a–f). We were unable to identify a *tf_{63}–smORF_{63}*-specific small-molecule inducer using the above compound screening strategy (Extended Data Fig. 4g). We conclude that the *Vibrio* 1F-97 polylysogen harbours two plasmid-like phages, each with SOS-independent pathways to lysis.

To examine the specificities of the two TF–smORF lysis pathways, we assessed whether either of the phage TFs could activate expression of the non-cognate *smORF* gene. Figure 2b shows that TF_{72} did not cause light to be produced from a P*smORF_{63}–lux* reporter, and TF_{63} did not drive light production from the P*smORF_{72}–lux* reporter. Notably, structural predictions of TF_{72} and TF_{63} using AlphaFold[32] suggested that the two TFs most closely resemble each other (root mean square deviation (RMSD) = 1.3 Å) (Extended Data Fig. 5a). They also resemble restriction modification controller proteins (RMSD = 0.5–0.9 Å) and the helix-turn-helix TF ClgR from *Mycobacterium smegmatis* (RMSD = 1.0–2.5 Å) (Extended Data Fig. 5b). AlphaFold modelling pinpointed the residues in each phage TF that constitute the DNA-recognition helix α3 (Extended Data Fig. 5a). Exchange of five residues within the recognition helix and flanking carboxy-terminal loop reversed the TF_{72} and TF_{63} preferences for *smORF_{72}* and *smORF_{63}* promoter DNA (the chimeric proteins are called TF_{72-5x} and TF_{63-5x}, respectively; Fig. 2b and Extended Data Fig. 3a). Thus, promoter specificity is conferred by a minimal set of non-contiguous unique residues in each TF. Consistent with the strict specificities of the phage pathway components, neither smORF, when produced together with the non-cognate cI repressor (pTet–*smORF_{63}* with *cI_{72}*-P_{R72}–*lux*; pTet–*smORF_{72}* with *cI_{63}*-P_{R63}–*lux*), generated P_{R}-driven

light production (Fig. 2c). Thus, each phage-encoded TF–smORF module is specific at two levels: *smORF* expression and smORF-driven repressor inactivation.

## QS controls TF–smORF modules

Our finding that phage genes encoding TF–smORF modules are not restricted to *repA–telN* genomic regions motivated us to expand our search. Preliminary BLAST searches revealed putative $TF_{63}$ homologues in 117 predicted phage genomes, which were almost exclusively associated with the *Vibrionaceae* family (Supplementary Table 2). In this dataset, we identified $TF_{63}$ homologues in four major genomic contexts: (1) in *repA–telN* loci; (2) within 5 kb of the partitioning gene *parB*; (3) in a heterogeneous genomic region containing small, variable ORFs of unknown function and; (iv), most rarely, within 10 kb of genes encoding predicted terminase and portal proteins. We used these findings to inform a second, guilt-by-association search for putative lysis-controlling modules that lack detectable sequence homology to $TF_{63}$ beyond those encoded by *Vibrionaceae* revealed in the BLAST search.

The NCBI database was searched for sequences encoding homologues of ParB and the major capsid protein from phage 63. We chose to examine the *parB*-associated loci more deeply because this genomic context was one of the most common in our preliminary $TF_{63}$ BLAST search and contained a conserved indicator gene (*parB*), which other common context groups lacked (Supplementary Table 2). The ParB search was anchored against a phage capsid protein to enrich for phage-associated *parB* loci instead of homologous sequences in plasmids or bacterial genomes. Capsid genes represent some of the most conserved viral sequences and were common among the genomes containing *parB* in our original set of 117 phages with $TF_{63}$ homologues[33]. We filtered the output of this search for hits encoding an apparent TF within 10 kb of the predicted *parB* gene given the close association between $TF_{63}$ and ParB in our earlier query. This search strategy revealed 56 distinct contigs (Supplementary Table 3), approximately 90% of which (50 out of 56) were putative linear plasmid-like phages based on detectable *repA–telN* loci. All 56 contigs were predicted to originate from phage genomes when analysed using VIBRANT, a hybrid machine-learning and protein similarity-based tool for phage identification[34] (Supplementary Table 1).

In one *par*-associated node of interest containing five phages, each phage carried an operon containing genes encoding an XRE-like DNA-binding protein and a LuxR-type TF (Fig. 3a and Extended Data Fig. 6a). LuxR-type proteins contain amino-terminal acyl homoserine lactone (HSL) AI-binding domains and C-terminal helix-turn-helix DNA-binding domains[35]. The *luxR* genes for two of the identified phages (ARM81ld of *Aeromonas* sp. ARM81, and Apop of *Aeromonas popoffii*) were associated with host strains that we had previously identified in a bioinformatics search for phage-encoded LuxR proteins[36]. In that work, we showed that the phage LuxR proteins could bind the HSLs *Aeromonads* are known to produce. However, we were unable to deduce the functions of the phage LuxR proteins at the time owing to an inability to obtain an *Aeromonas* sp. ARM81 isolate with the phage and the genetic intractability of *A. popoffii*. To expand our understanding of the roles of these phage modules, we successfully obtained *Aeromonas* sp. ARM81 and we developed genetic tools to study *A. popoffii*.

First, we investigated the *A. popoffii* XRE–LuxR ($XRE_{Apop}$–$LuxR_{Apop}$) pair. An unannotated 156 nucleotide *smORF* resided approximately 150 nucleotides upstream of the $xre_{Apop}$–$luxR_{Apop}$ operon near the *par* locus (Extended Data Fig. 6a). To determine whether it is regulated by $XRE_{Apop}$–$LuxR_{Apop}$, a plasmid carrying a Tc-inducible $xre_{Apop}$–$luxR_{Apop}$ was transformed into *E. coli* harbouring *lux* fused to the promoter of the candidate *smORF* ($PsmORF_{Apop}$–*lux*). Light production from $PsmORF_{Apop}$–*lux* commenced only when the *E. coli* reporter strain was supplied with C4-HSL (Extended Data Fig. 6b), an AI natively produced

by *Aeromonads*[37]. Thus, $XRE_{Apop}$ and the $LuxR_{Apop}$–AI complex control *smORF* expression. Consistent with this finding, deletion of the *smORF* locus from Apop abolished $XRE_{Apop}$–$LuxR_{Apop}$-AI-dependent lysis of *A. popoffii* but not ciprofloxacin-dependent lysis (Extended Data Fig. 6c). Analogous to what we show in Fig. 1d and Extended Data Figs. 1g and 4b,f, the Apop cI protein $cI_{Apop}$ shifted $P_{R\text{-}Apop}$ DNA in vitro (Extended Data Fig. 6d), and $smORF_{Apop}$ inactivated $cI_{Apop}$ (Extended Data Fig. 6e). Lastly, unlike DNA damage, $smORF_{Apop}$ did not lead to $cI_{Apop}$ autoproteolysis (Extended Data Fig. 6f). The *Aeromonas* sp. ARM81 phage ARM81ld XRE–LuxR module functioned analogously to the *A. popoffii* XRE–LuxR module (Extended Data Fig. 7a–e). Thus, these XRE–LuxR-controlled modules likely act as orthogonal SOS-independent pathways to induction.

The Apop and ARM81ld phage TF–smORF modules are distinctive among the characterized set of TF–smORF modules given that two TFs (XRE and LuxR) are apparently involved in the activation of *smORF* expression. To define the individual and combined contributions of $XRE_{Apop}$ and $LuxR_{Apop}$, we generated expression vectors carrying only pTet–$xre_{Apop}$ or only pTet–$luxR_{Apop}$ and tested them in the $PsmORF_{Apop}$–*lux* assay. Neither $XRE_{Apop}$ nor $LuxR_{Apop}$ alone drove reporter output irrespective of the presence of C4-HSL. This result indicated that in addition to AI, $smORF_{Apop}$ expression depends on both TFs (Fig. 3b). To validate these findings, we engineered $xre_{Apop}$–$PsmORF_{Apop}$–*lux* and $luxR_{Apop}$–$PsmORF_{Apop}$–*lux* reporter plasmids, which carry the identical $PsmORF_{Apop}$–*lux* reporter sequence but include either full-length $xre_{Apop}$ or full-length $luxR_{Apop}$ controlled by the native promoter. We introduced pTet–$xre_{Apop}$ or pTet–$luxR_{Apop}$ into *E. coli* harbouring the reporters. Heterologous $xre_{Apop}$ expression induced light production only when $luxR_{Apop}$ was present on the reporter plasmid and AI was supplied (Fig. 3c,d). By contrast, heterologous $luxR_{Apop}$ expression did not induce light production in either case, irrespective of the presence of AI (Fig. 3c,d). We interpret these results as follows: $XRE_{Apop}$ activates the expression of the $xre_{Apop}$–$luxR_{Apop}$ operon, driving both $XRE_{Apop}$ and $LuxR_{Apop}$ production. Together, $LuxR_{Apop}$ bound to C4-HSL and $XRE_{Apop}$ are required to activate $smORF_{Apop}$ expression, which induces the phage lytic cascade and host-cell lysis (Fig. 3e). Confirming the proposed regulatory arrangement in vivo, quantitative PCR with reverse transcription (RT-qPCR) results revealed that overexpression of $xre_{Apop}$ in *A. popoffii* led to the rapid activation of expression of $xre_{Apop}$ and $luxR_{Apop}$, whereas overexpression of $luxR_{Apop}$ did not (Extended Data Fig. 7f).

## Induction dictates phage competition

Inter-prophage competition is predicted to arise in polylysogens when co-occurring prophages are induced by the same trigger (for example, DNA damage)[23]. Our above discovery of non-canonical phage lysis pathways established the possibility that the outcomes of such competitions could depend on the particular inducer that activates lysis. *Vibrio* 1F-97 cells lyse following phage induction by ciprofloxacin (DNA damage), production of $smORF_{63}$ or production of $smORF_{72}$ (Fig. 1c and Extended Data Fig. 4e). Nonetheless, the amounts of phage 63 and phage 72 virions released under each condition could reflect the consequences of inter-prophage competition. Whole-genome sequencing of phage particle preparations from *Vibrio* 1F-97 cultures induced by ciprofloxacin, $smORF_{63}$ or $smORF_{72}$ revealed that ciprofloxacin treatment drove the production of comparable quantities of DNA corresponding to both phage 72 and phage 63 (Fig. 4a). By contrast, cultures induced by $smORF_{72}$ or $smORF_{63}$ resulted in near-exclusive production of phage 72 particles and phage 63 particles, respectively (Fig. 4a). Thus, under non-SOS-inducing conditions, the phage produced depends on the smORF driving lysis.

To date, investigations of prophage induction in polylysogenic hosts have been limited to population-level measurements, as in Fig. 4a. Consequently, it is unclear whether in a single host bacterium, co-residing prophages can be simultaneously induced and acquire the

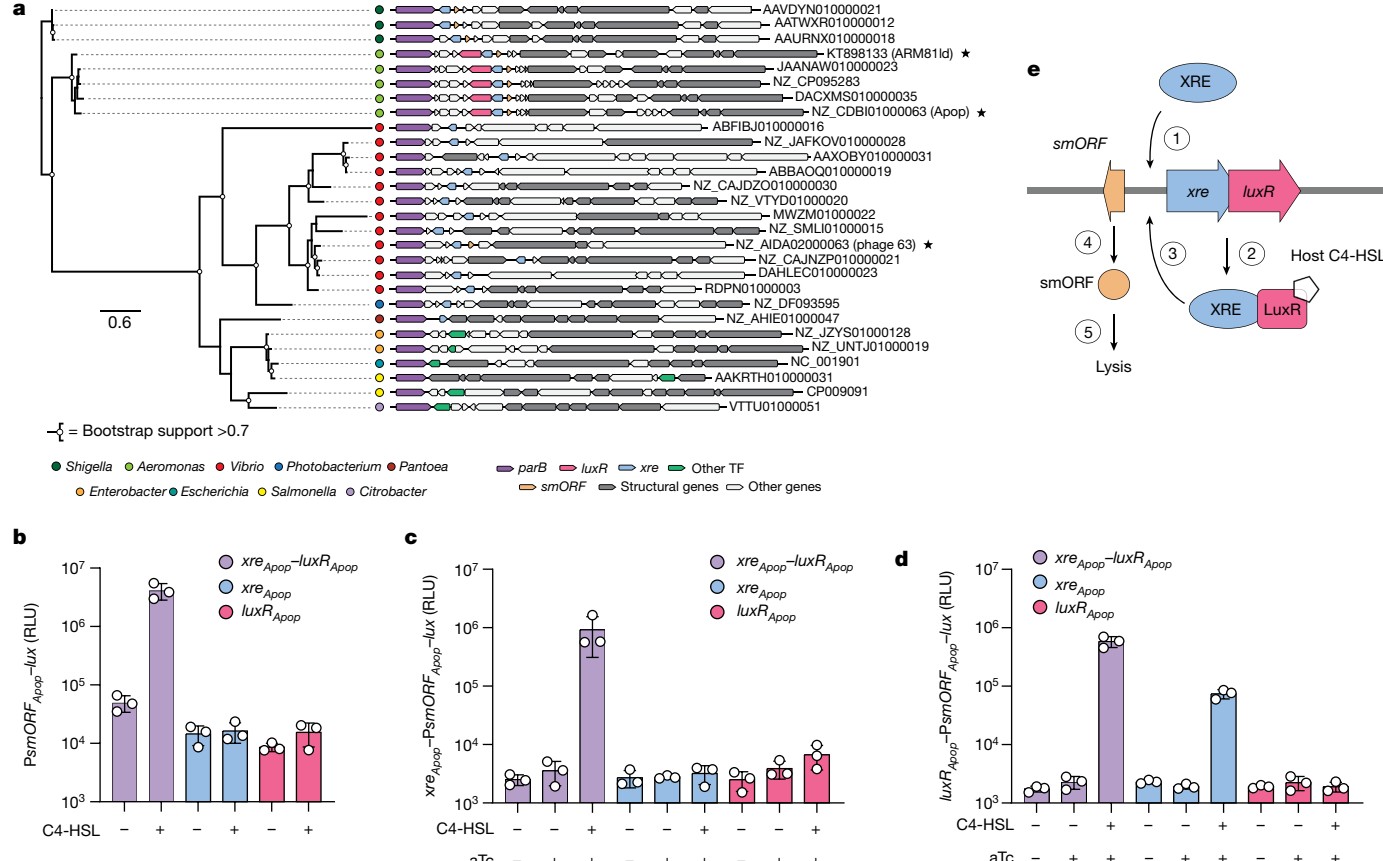

**Fig. 3 | Activation of P*smORF*<sub>Apop</sub> requires XRE<sub>Apop</sub>, LuxR<sub>Apop</sub> and the C4-HSL AI. a**, Phylogenetic tree (left) of 28 representative ParB proteins. Gene neighbourhoods (right) for the 28 loci encoding *parB* (purple) and the major capsid protein (grey). ARM81ld, Apop and phage 63 are denoted with stars. Genes encoding predicted TFs that are not *luxR* or *xre* are in green. NCBI accession numbers are depicted to the right of each sequence. Scale bar as in Fig. 1a. **b**, P*smORF*<sub>Apop</sub>–*lux* activity from a plasmid in *E. coli* carrying a second vector with the designated aTc-inducible gene (or genes). All media contained aTc. Supplementation with DMSO or C4-HSL is denoted by minus and plus symbols, respectively. **c**, P*smORF*<sub>Apop</sub>–*lux* activity from a plasmid in *E. coli* carrying *xre*<sub>Apop</sub> under its native promoter and a second vector with the designated aTc-inducible gene (or genes). Media contained the indicated combinations of water (−) or aTc (+), and DMSO (−) or C4-HSL (+). **d**, P*smORF*<sub>Apop</sub>–*lux* activity from a plasmid in *E. coli* carrying *luxR*<sub>Apop</sub> under its native promoter and a second vector with the designated aTc inducible gene (or genes). Treatments as in **c**. **e**, Proposed model for the regulation of the TF–smORF module in Apop. (1) XRE<sub>Apop</sub> activates the expression of the *xre*<sub>Apop</sub>–*luxR*<sub>Apop</sub> operon, and (2) increased production of LuxR<sub>Apop</sub> and XRE<sub>Apop</sub> occurs. (3) LuxR<sub>Apop</sub> when bound to the C4-HSL AI ligand (pentagon), together with XRE<sub>Apop</sub>, activates expression of the counter-oriented *smORF*<sub>Apop</sub> gene, and activates the expression of *xre*<sub>Apop</sub>–*luxR*<sub>Apop</sub>. (4) smORF<sub>Apop</sub> inhibits the cI<sub>Apop</sub> repressor, (5) leading to host-cell lysis. The mechanism underlying the co-requirement for AI-bound LuxR<sub>Apop</sub> and XRE<sub>Apop</sub> to activate P*smORF*<sub>Apop</sub> remains to be determined. Data are presented as the mean ± s.d. with *n* = 3 biological replicates (**b**–**d**). RLU as in Fig. 1d (**b**–**d**).

resources needed for replication or, alternatively, whether only one of the co-residing prophages successfully replicates. Specifically, in the case of the *Vibrio* 1F-97 polylysogen, multiple mechanisms could produce the mixture of phage 72 and phage 63 particles that are produced following DNA damage. First, each host cell could produce both phage 72 and phage 63 particles. Second, each host cell could produce exclusively phage 72 particles or exclusively phage 63 particles. Third, a subset of host cells could produce exclusively phage 72 particles, another subset could produce exclusively phage 63 particles and a final subset could produce both phage 72 and phage 63 particles. To determine which possibility is correct, we used single-molecule RNA fluorescent in situ hybridization (smRNA-FISH) to mark and visualize phage induction in individual cells of the *Vibrio* 1F-97 polylysogen. Our strategy relied on the use of probes specific for phage 72 or for phage 63 genes that are expressed only during entry into the lytic cycle (Methods). Figure 4b shows that following DNA damage, a subset of cells (33%) exclusively harboured RNA from phage 72, a subset (9%) harboured RNA from phage 63 and a subset (28%) harboured RNA from both phages. Thus, DNA damage causes heterogeneity in phage induction at the single-cell level. By contrast, and as predicted on the

basis of results presented in Fig. 4a, only phage-72-specific RNA was detected in cells induced by smORF<sub>72</sub>, and only phage-63-specific RNA was detected in cells induced by smORF<sub>63</sub> (Fig. 4b). These findings indicate that activation of TF–smORF modules drive exclusivity in prophage induction at the single-cell level. Conversely, SOS activation leads to multiple populations of differentially induced cells, including those expressing the lytic genes of both phages and of either phage alone.

Previous work had determined that *Aeromonas* sp. ARM81 is polylysogenic and contains an integrated prophage, ARM81mr, in addition to the plasmid-like ARM81ld prophage[38]. We have no genomic evidence for an additional sensory pathway on the ARM81mr phage. We used the identical single-cell imaging smRNA-FISH analysis described above to discover the mechanism that gives rise to the sets of released phages from *Aeromonas* sp. ARM81 following induction with either ciprofloxacin or smORF<sub>ARM81ld</sub>. The results mirrored those for *Vibrio* IF-97. DNA damage resulted in cells expressing lytic genes of exclusively phage ARM81mr (36%), exclusively phage ARM81ld (13%) or both phages (46%). Conversely, smORF<sub>ARM81ld</sub> production drove the expression of only phage ARM81ld (Fig. 4c). Indeed, quantitation of the virions released following treatment with ciprofloxacin or following induction

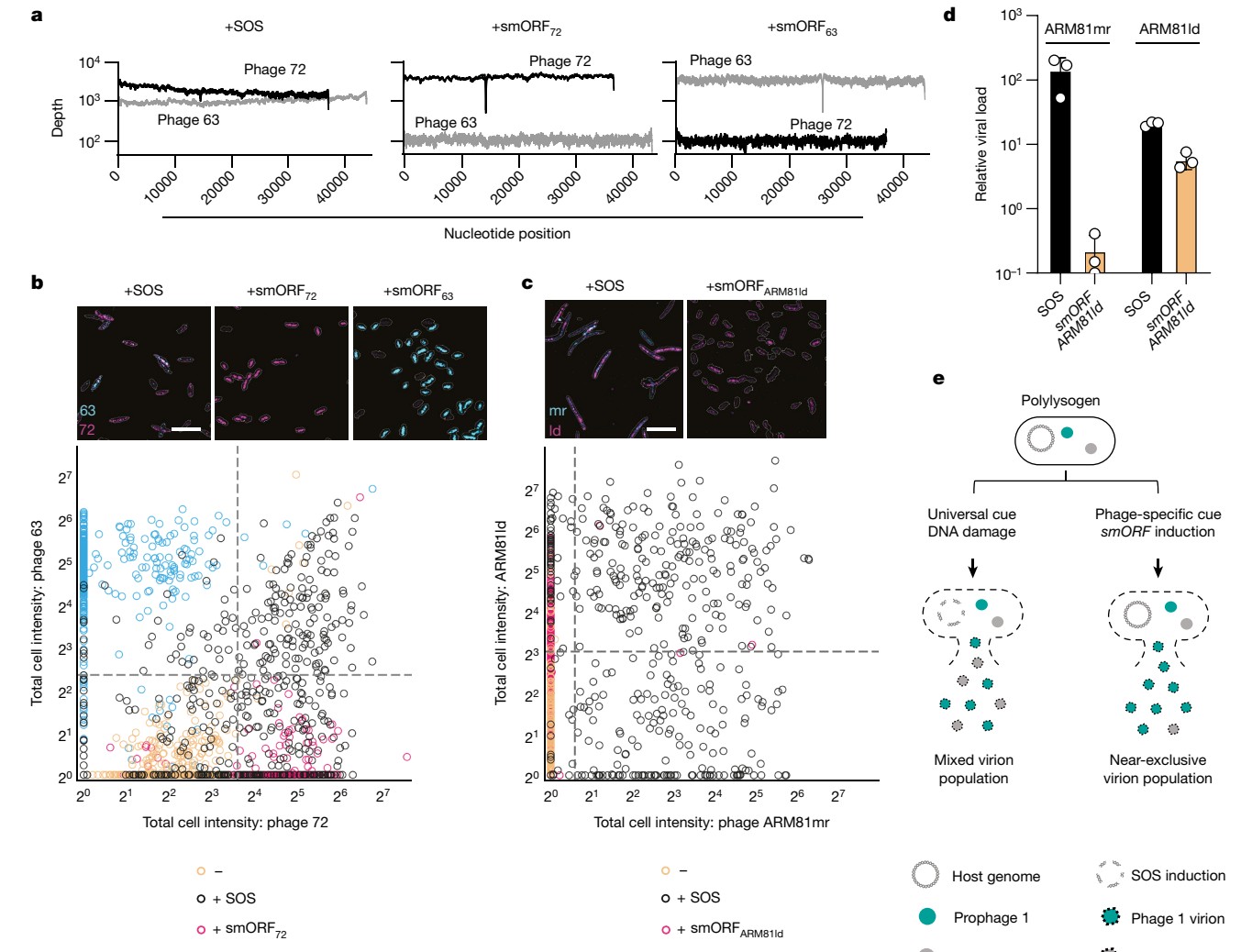

**Fig. 4 | DNA damage drives discrete subsets of cells producing one, the other or both phages whereas TF–smORF modules exclusively drive production of only the phage encoding that module in polylysogenic hosts. a**, Sequencing of viral particles prepared from *Vibrio* 1F-97 cultures carrying an empty vector, aTc-inducible *smORF$_{72}$* or aTc-inducible *smORF$_{63}$* grown in medium containing ciprofloxacin and aTc. Depth refers to the number of read counts. **b**, Top, representative smFISH images of phage 63 and phage 72 lytic genes in *Vibrio* 1F-97 cells following SOS activation or *smORF* induction as in **a**. Images are maximum *z*-projections of raw smFISH fluorescence for phage 72 (72; magenta) and phage 63 (63; cyan) genes with cells outlined in white. Bottom, phage 72 versus phage 63 smFISH intensity per *Vibrio* 1F-97 cell in the absence of phage induction (orange, *n* = 677 cells), with SOS activation (black, *n* = 516 cells), *smORF$_{72}$* induction (magenta, *n* = 375 cells) or *smORF$_{63}$* induction (cyan, *n* = 485 cells). Uninduced cells were used to determine boundaries (grey dotted lines; Methods) to delineate cells displaying no phage induction (bottom left quadrant), exclusively phage 72 induction (bottom right quadrant), exclusively phage 63 induction (top left quadrant), or both phage 63 and phage 72 induction (top right quadrant). Scale bar, 10 µm. **c**, Top, representative smFISH

images of phage ARM81mr and phage ARM81ld early lytic gene expression in *Aeromonas* sp. cells following SOS activation or *smORF$_{ARM81ld}$* induction as in **d**. Bottom, images as in **b** for phage ARM81mr (mr; cyan) and phage ARM81ld (ld; magenta) genes. Phage ARM81mr versus ARM81ld smFISH intensity per *Aeromonas* sp. cell in the absence of phage induction (orange, *n* = 516 cells), with SOS activation (black, *n* = 463 cells) or *smORF$_{ARM81ld}$* induction (magenta, *n* = 477 cells). Scale bar and quadrants as in **b** (Methods). **d**, Detection of viral particles from *Aeromonas* sp. ARM81 carrying aTc-inducible *smORF$_{ARM81ld}$* grown in medium containing ciprofloxacin (black) or aTc (orange). Relative viral load is the amount of ARM81mr-specific or ARM81ld-specific DNA (*cl$_{ARM81mr}$* and *cl$_{ARM81ld}$*, respectively) in the induced samples relative to an uninduced sample measured by qPCR. **e**, Model for how TF–smORF modules influence inter-prophage competition. Left, a ubiquitous prophage inducer promotes competition because multiple prophages share host cell resources. Right, prophage-specific induction through a TF–smORF leads to exclusivity because only the induced phage garners host cell resources. Data are presented as the mean ± s.d. with *n* = 3 biological replicates (**a**–**c**) or as the mean ± s.d. with *n* = 3 biological replicates and *n* = 4 technical replicates (**d**).

of *xre$_{ARM81ld}$–luxR$_{ARM81ld}$* expression produced results consistent with the smRNA-FISH analysis. Specifically, administration of ciprofloxacin led to a 140-fold increase in ARM81mr phage particles and a 22-fold increase in ARM81ld particles relative to when no inducer was added (Fig. 4d). By contrast, induction of expression of *xre$_{ARM81ld}$–luxR$_{ARM81ld}$* with C4-HSL led to a fivefold increase in ARM81ld particles, whereas ARM81mr particles were reduced by fivefold compared with their uninduced levels (amounting to a 700-fold reduction in ARM81mr particles compared

with ciprofloxacin treatment; Fig. 4d). Together, our results from *Vibrio* 1F-97 and *Aeromonas* sp. ARM81 demonstrate that DNA damage results in the induction of and competition between co-residing prophages, whereas possession of an additional sensory pathway drives host-cell lysis exclusively by the phage that encodes the module (Fig. 4e). Thus, the particular induction cue dictates the distribution of phage particles produced by the polylysogenic host. Moreover, single and dual phage production is possible from an individual host cell.

## Discussion

Despite most bacteria being predicted to be polylysogenic, an understanding of the consequences of polylysogeny on the hosts and on their resident phages has been constrained by the limited models available and a lack of known prophage induction cues beyond the SOS response. Currently, competition among co-residing temperate prophages is generally considered a sprint, whereby in response to a single trigger (that is, the SOS response), differences in particular phage properties (for example, replication rates, packaging rates, burst size, among others), dictate how many particles of each phage are produced. Our current work in polylysogens harbouring phages with multiple pathways to lysis shows that differential phage induction can occur, and which phage or phages are produced varies depending on the induction cue. Although tuning into the host SOS response remains a universal mechanism by which prophages perceive host-cell stress, the additional sensory pathways we discovered here suggest that more specialized conditions exist that favour the induction of one phage over another.

In this work, we focused on discovering and characterizing SOS-independent lysis induction pathways on linear plasmid-like phages that encode autoproteolytic cI repressors. We uncovered a pattern in which previously unknown phage regulatory components are encoded adjacent to genes characteristic of linear plasmid-like phages (that is, near replication and partition machinery genes). To make headway, we constrained our initial work to this subset of phages, which constitute a specific subtype present in Gram-negative bacteria. We predict that SOS-independent pathways to lysis are widespread among phages but may require different search strategies to reveal them. Indeed, RecA-independent phage induction has been shown to occur in lambdoid phages in *E. coli*; however the molecular mechanisms that underlie these alternative pathways are unknown[39].

Our single-cell imaging analyses further revealed that phage induction through the SOS cue drives a heterogenous outcome from polylysogens. That is, individual cells produce one type of phage, the other type of phage or a mixture of both phages. In the cases in which only one of the lytic genes of the phages is expressed in a host cell, we speculate that perhaps some stochastic process causes only one phage to enter the lytic cycle or, alternatively, one phage achieves a sufficient head start over the other phage and it monopolizes host resources, quenching production of the other phage. Our finding that a mixture of phages can be produced by a single host cell shows that multiple prophages can enter the lytic cycle and presumably compete for the host resources required to replicate and produce viral particles. Regarding *Aeromonas*, we note that although the smFISH analyses showed that the SOS cue drives similar levels of early lytic gene induction from both phages, phage ARM81mr ultimately outcompeted phage ARM81ld, producing tenfold more viral particles. Differences in rates of downstream processes (for example, replication and/or packaging) could drive this discrepancy. Our use of smRNA-FISH for imaging prophage induction in polylysogenic bacteria enabled this visualization of inter-prophage competition at the single-cell level. Going forward, this approach can be adapted to study other polylysogens and, for example, used for investigations of phage gene expression on rapid timescales during prophage induction or infection.

Regarding the molecular mechanisms that underlie inter-prophage competition, our results showed that induction of a TF–smORF module of a phage does not cross-activate the lytic programme of another co-resident prophage. In *Vibrio* 1F-97, this specificity was achieved at two levels: specific TF-mediated activation of *smORF* expression and specific smORF-driven repressor inactivation. AlphaFold modelling of the TFs revealed similarity in the global folds of the two proteins; however, the specificity for their respective target promoters is conferred by a few key residues within the DNA-recognition helix. In contrast to our findings, investigation of polylysogenic *Salmonella* strains revealed that some phage antirepressor proteins can inactivate cognate

and non-cognate repressors, thereby enabling the synchronization of induction of multiple prophages[40]. This arrangement is thought to be vital for prophages with slow induction responses, allowing them to piggyback off of prophages that are more rapidly induced[40]. The finding that prophage induction by an additional cue does not trigger induction of co-residing prophages could be a strategy that proves especially successful when host resources are limiting because it ensures exclusive reproduction and dissemination of only the induced phage.

The two phage-encoded LuxR proteins investigated here required partner XRE proteins to activate the expression of their counter-oriented *smORF* genes, a requirement that does not exist for most bacterial LuxR QS receptors. One recent example, however, demonstrates that the activity of the *Pseudomonas aeruginosa* LuxR receptor, called RhlR, is modulated by direct interaction with the PqsE protein[41]. Whether the phage LuxR and XRE proteins interact directly, and if so, if the mechanism parallels that for RhlR–PqsE, remains to be determined. Finally, we found that *xre* and *luxR* do not always co-occur in phage genomes. A tBLASTn search using XRE$_{Apop}$ as the query revealed three contigs, likely from linear plasmid-like phages, in different *Shigella sonnei* genomes (Extended Data Fig. 6a). Unlike in the *Aeromonas* phages, these putative *Shigella* phages lack *luxR* genes. *Shigella* are not known to produce HSL AIs, thus a LuxR component might not provide benefits to the phage in the *Shigella* host. It is possible that particular sets of regulatory components encoded on different plasmid-like phages have been shaped through evolution, presumably by relevant host-sensory cues.

The genes encoding the phage TFs TF$_{72}$, TF$_{63}$, XRE$_{Apop}$–LuxR$_{Apop}$ and XRE$_{ARM81ld}$–LuxR$_{ARM81ld}$ all required synthetic induction to drive lysis, a situation that mirrors our findings for VqmA$_{Phage}$ in phage VP882 (ref. 19). Although not investigated here, it is possible that these regulators are produced in infected cells and they function to activate lysis before any additional newly infecting phages can establish lysogeny. Determining the roles of these modules in all stages of the phage life cycle is a focus of our ongoing work. Thus far, we have not succeeded in identifying the natural inducers of these TF–smORF cascades. However, we note that DNA-damaging compounds, the universally used phage inducers, do not generally occur in nature at the concentrations used in experiments. Thus, the identities of the natural cues that drive lysis even in intensively studied (that is, SOS-dependent) prophages remain unknown. Presumably, the signals that induce the additional pathways to lysis revealed here, as well as others that are identified going forward, could be particular to each host–phage partnership and the niche in which they reside. We propose that although discovered long after the SOS cue, some or all of these TF–smORF pathways may be key determinants of phage lifestyle transitions and the arbiters of inter-prophage competition in real-world settings.

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

## Methods

### Sequence retrieval, RepA–TelN and ParB-associated loci

To identify examples of phage genomes with convergently oriented *telN* and *repA* genes, we examined sequences from the following databases: NCBI nt, IMG/VR v.3 (specifically the file IMGVR_all_nucleotides.fna)[28], Cenote Human Virome Database v.1.1 (CHVD_clustered_mash99_v1.fna)[24], Global Ocean Virome 2 database (GOV2_viral_populations_larger_than_5Kb_or_circular.fna)[42], the Gut Phage Database (GPD_sequences.fna)[25], a curated set of linear plasmid-phages (retrieved through NCBI accession numbers in table S4 of ref. 26) and the Metagenomic Gut Virus Catalog (mgv_contigs.fna)[29]. In February 2022, most databases were searched using both tBLASTn and profile-HMM-based search strategies (the exception being NCBI nt, which was too large to annotate anew using profile HMMs). A manual examination of established lysis-control loci[19] revealed that the associated *telN* gene always encoded a protein that matched closely to Pfam profile PF16684, whereas the following set of seven families encompassed the diversity of observed *repA* sequences: PF13362, PF02399, PF08707, PF10661, PF02502 and PF13604. Thus, we extracted these seven families from Pfam (v.34)[43] into a custom HMM database. We then used the gene finder MetaGeneMark[9] to predict ORFs from the aforementioned nucleotide files using default parameters. The resulting proteins were used in a profile HMM search with HMMER3 (ref. 44) against the *repA*/*telN* custom database, and nucleotide sequences were extracted that contained both *telN* and *repA* genes only if they were convergently oriented and within 10 kb of one another. To complement this search strategy, we also used the predicted TelN and RepA proteins from vibriophage VP882 in a tBLASTn search against these nucleotide databases (NCBI accession numbers YP_001039865 and YP_001039868, respectively). We retained only the hits with *e*-values better than 0.001, which also covered >50% of the query sequence. For consistency, we re-annotated all retrieved sequences using a common method, as follows. We used the gene-finder MetaGeneMark[45] to predict ORFs using default parameters. We next used their amino acid sequences in a profile HMM search with HMMER3 (ref. 44) against TIGRFAM[46] and Pfam[43] profile HMM databases. The highest scoring profile was used to annotate each ORF. As above, we further refined our database by considering only contigs with convergently oriented *telN* and *repA* genes within 10 kb of one another. The HMM-based and tBLASTn-based search strategies produced highly redundant (but not identical) sequence sets, as the same databases were used and many identical sequences are listed across multiple databases. Thus, we dereplicated our combined sequence files using cd-hit-est with the following parameters: '-c 1.0 -aS 1.0 -g 1 -d 0'. We manually examined the DNA sequence located between each *telN* and *repA* gene from this dereplicated set, extracting the intervening nucleotides to a second locus-specific dataset. We dereplicated these sequences as described above to produce the final set of 274 loci referenced in the text and detailed in Supplementary Table 1. The NCBI nucleotide accession that encodes phage 63 (NZ_AIDA02000063) is dated 20 March 2022, after our initial February search. The sequence for phage 72 appears more than once in the NCBI nucleotide database (NZ_AIDA02000072, dated 20 March 2022, and KP795532, dated 18 November 2019). Presumably, these similar entries result from separate sequencing analyses of identical (or similar) bacteria. For these reasons, our February BLAST search only initially revealed phage 72, whereas phage 63 was subsequently discovered.

To investigate phage genomes for potential $TF_{63}$-associated marker genes, we first performed a preliminary BLASTp search against the NCBI nr/nt database using $TF_{63}$ as a query (accession number WP_016786069). We filtered the top 5,000 hits from this search for those with over 35% amino acid identity across 70% of the query sequence and retrieved the corresponding nucleotide file in NCBI, retaining the DNA ±50 kb from the gene boundaries. This analysis produced 744 sequences that were then filtered for those of predicted phage origin using the phage prediction tool VIBRANT[34] (v.1.2) with default parameters. VIBRANT indicated that 200 sequences in this set were putatively phage-derived. Some of these putative phage $TF_{63}$ sequences aligned poorly to our original query, so a second tBlastn search was performed using the original $TF_{63}$ protein. From the 200 putative phage genomes, only those with tBlastn hits that had an *e*-value better then $e^{-10}$ were retained. Finally, we removed one short sequence (<5 kb) and dereplicated the remaining phage sequences at 95% nucleotide identity with cd-hit-est using the following parameters: '-c 0.95 -aS 0.95 -g 1 -d 0'. This strategy produced 117 sequences that were then uniformly annotated as described above and manually categorized into the genomic context groups described in Supplementary Table 2. The phage prediction tool VIBRANT was also used to assign a likely phage origin to the sequences listed in Supplementary Table 1, as described above.

To identify examples of putative lysis-control loci associated with the *parB* gene, we again performed a BLASTp search against the NCBI nr/nt database, this time using the ParB protein from phage 63 as a query (accession number WP_016786072). From the top 10,000 proteins revealed by this search, we retrieved the corresponding nucleotide file in NCBI and examined the locus surrounding *parB* by downloading DNA ±50 kb from the gene boundaries (this strategy retrieved the full contig sequence for many of the associated nucleotide files). To identify *parB* genes associated with the phage structural locus present in phage 63, we used tBLASTn to query these sequences with the major capsid protein from this phage (accession number WP_016786053), which is one of the most conserved phage protein folds[47]. We filtered for hits with over 25% amino acid identity across 70% of the query sequence, revealing 121 putative *parB* loci present on 118 distinct sequences. These 121 loci were dereplicated as described for *telN* and *repA* and filtered for the presence of a full-length *parB* gene and a predicted TF within 10 kb of *parB*. This analysis produced 56 sequences, which are referenced in Supplementary Table 3.

### Phylogenetic analysis of TelN and ParB

From a manual examination of the retrieved RepA-encoding and TelN-encoding sequences, we observed that all TelN proteins shared the same profile HMM as their top-scoring annotation (PF16684.8). This pattern stood in contrast to predicted RepA-encoding sequences, which hit best to one of six different protein families. The consistent TelN annotation indicated to us that this protein may be conserved despite extensive sequence divergence among the phage genomes considered, and thus, it provided a good phylogenetic bellwether for this locus. Indeed, the set of non-redundant, full-length TelN proteins from these datasets aligned well using MUSCLE with default parameters[48]. A preliminary phylogenetic tree was generated from these sequences using the Geneious Tree Builder per the UPGMA clustering method and with a Jukes–Cantor distance model. From this tree, a subset of 34 sequences was selected to best represent the full TelN phylogenetic tree and the genetic diversity encoded by its associated gene neighbourhoods. Figure 1a depicts a phylogenetic tree generated from these 34 representative sequences, which was produced using PhyML with the LG substitution model and bootstrapped 100 times. A similar procedure was used to produce the ParB phylogeny depicted in Fig. 3a. In this case, a preliminary phylogenetic tree was produced from the set of 56 ParB proteins in the full dataset using the Geneious Tree Builder per the UPGMA clustering method and with a Jukes–Cantor distance model. From this tree, a subset of 28 sequences was selected to best represent the full ParB phylogenetic tree and the genetic diversity encoded by its associated gene neighbourhoods. This final ParB tree was produced using PhyML with parameters identical to those used for TelN.

### Prediction of autoproteolytic cI proteins

All 271 predicted cI proteins were used to perform a batch search of the NCBI Conserved Domain (CD) database. The output of this search

included conserved domains and other protein features for each cI protein queried. Among these features were predicted catalytic sites that corresponded to known catalytic residues in canonical autoproteolytic cI proteins[14]. Thus, we used this feature table to define the 61 predicted autoproteolytic cI proteins and labelled the remaining 210 proteins as putatively non-autoproteolytic. The clustering analysis presented in Supplementary Table 1 (summary statistics tab) was performed using the DNA sequences between *repA* and *telN* genes and grouped sequences that shared 80% nucleotide identity over 95% of the length of the shorter sequence in the same cluster. We used the program 'cd-hit-est' with the following parameters: -c 0.8 -aS 0.95 -g 1 -d 0.

### Phylogenetic analysis of linear plasmid-like phages encoding *par*-associated *xre* modules in *Shigella* and *Aeromonas*

NCBI BLASTp of the DNA sequence encoding XRE$_{Apop}$ was used as a representative of the three *Aeromonas* phages identified in the *par*-associated analysis (detailed above) retrieved three predicted linear plasmid-like phages in *Shigella*. For each phage genome, genes were called using Prodigal (v.2.6.3)[49], and gene diagrams were constructed. Genes were annotated using Prokka (v.1.11)[50], and annotations were supplemented with NCBI BLASTp searches by hand[51]. The phage genome tree (Extended Data Fig. 3a) was constructed using VICTOR[52], producing an average support of 78%. The numbers above branches are pseudo-bootstrap support values from 100 replications.

### Bacterial strains and growth conditions

*E. coli* and *Aeromonads* were grown with aeration in Luria–Bertani (LB-Miller, BD-Difco) broth. *Vibrio* strains were grown in LB medium with 3% NaCl. All strains were grown at 30 °C. Strains used in the study are listed in Supplementary Table 4. Unless otherwise noted, antibiotics, were used at the following concentration: 100 µg ml$^{-1}$ ampicillin (Sigma), 50 µg ml$^{-1}$ kanamycin (GoldBio) and 5 µg ml$^{-1}$ chloramphenicol (Sigma). Inducers were used as follows: *E. coli*: 200 µM isopropyl β-D-1-thiogalactopyranoside (IPTG, GoldBio) and 50 ng ml$^{-1}$ aTc (Clontech); *Vibrios*: 500 ng ml$^{-1}$ ciprofloxacin (Sigma) and 50 ng ml$^{-1}$ aTc; *Aeromonads*: 1 µg ml$^{-1}$ ciprofloxacin and 5 ng ml$^{-1}$ aTc. C4-HSL was supplied at a final concentration of 10 µM except for the experiment shown in Fig. 3b, in which it was administered at specified doses, as indicated in the figure.

### Cloning techniques

All primers and dsDNA (gene blocks) used for plasmid construction, qPCR and electrophoretic mobility shift assays (EMSAs) listed in Supplementary Table 5 were obtained from Integrated DNA Technologies or Twist Bioscience. Gibson assembly, intramolecular reclosure and traditional cloning methods were used for all cloning. PCR with iProof was used to generate insert and backbone DNA. Gibson assembly relied on the HiFi DNA assembly mix (NEB). The Apop-based and ARM81ld-based P$_R$–*lux* reporter constructs in *E. coli* (JSS-3346k and JSS-3348k, respectively) required the addition of a second copy of the *cI* repressor under its native promoter in the plasmid for function (Supplementary Table 5). All enzymes used in cloning were obtained from NEB. Plasmids used in this study are listed in Supplementary Table 6. Transfer of plasmids into *Vibrio* 1F-97, *A. popoffii*, and *Aeromonas* sp. ARM81 was carried out by conjugation followed by selective plating on TCBS agar supplemented with kanamycin, LB plates supplemented with ampicillin and kanamycin, and LB plates supplemented with kanamycin and chloramphenicol, respectively.

### Growth, lysis and reporter assays

Overnight cultures were back-diluted 1:100 with fresh medium with appropriate antibiotics before being dispensed (200 µl) into 96-well plates (Corning Costar 3904). Cells were grown in the plates for 90 min before ciprofloxacin, aTc or C4-HSL was added as specified. Wells that did not receive treatment received an equal volume of water or DMSO.

A BioTek Synergy Neo2 Multi-Mode reader was used to measure OD$_{600}$ and bioluminescence. RLUs were calculated by dividing the bioluminescence readings by the OD$_{600}$ at that time.

### RT–qPCR

Overnight cultures were back-diluted 1:100 and grown for 90 min before administration of aTc or ciprofloxacin at the indicated concentrations. In Extended Data Figs. 1d and 7f, cells were collected at $T = 0$ and 15 min, and in Extended Data Fig. 1h, at $T = 0$ and 90 min following induction. Collected cells were treated with RNAProtect Bacteria Reagent (Qiagen) according to the supplier's protocol. Total RNA was isolated from cultures using an RNeasy Mini kit (Qiagen). RNA samples were treated with DNase using a TURBO DNA-free kit (Thermo). cDNA was prepared as described using SuperScriptIII Reverse Transcriptase (Thermo). SYBR Green mix (Quanta) and an Applied Biosystems QuantStudio 6 Flex Real-Time PCR detection system (Thermo) were used for real-time PCR. Each cDNA sample was amplified in technical quadruplicate and data were analysed by a comparative CT method, in which the indicated target gene was normalized to an internal bacterial control gene (*rpoB*).

### qPCR and viral preparation

Viral preparations consisted of non-chromosomal DNA (RQ1, RNase-Free DNase, Promega) prepared from 1 ml of cells of the indicated strains. Overnight cultures were back-diluted 1:100 and grown for 90 min before being divided into 3 equal volumes and exposed to treatments as specified. Cultures were grown for an additional 5 h before collection of cell-free culture fluids (Corning SpinX). qPCR reactions were performed as described above for RT–qPCR assays. Next, 1 µl of purified non-chromosomal DNA was used for each qPCR reaction. Data were analysed by normalizing the CT values of samples treated with ciprofloxacin or aTc to the CT values of samples treated with water using a primer set to the indicated phage. Viral preparations for whole-genome sequencing (SeqCenter) were prepared exactly as described for qPCR, except by column purification (Phage DNA Isolation Kit, Norgen Biotek).

### Small-molecule screening for P*smORF$_{72}$*–*lux* and P*smORF$_{63}$*–*lux* inducers in *Vibrio* 1F-97

Overnight cultures of *Vibrio* 1F-97 strains harbouring a plasmid encoding either P*smORF$_{72}$*–*lux* or P*smORF$_{63}$*–*lux* were back-diluted 1:100 and grown for 90 min before being dispensed into 23 × 96-well Biolog plates (PM3b-PM25) and 3 × 96 well plates of curated compounds from the Seyedsayamdost Group (Princeton). Plates were incubated at 30 °C overnight before measurement of OD$_{600}$ and bioluminescence. Biolog conditions or library compounds that inhibited growth were retested in a *Vibrio* 1F-97 strain harbouring a reporter for phage lambda induction (*cI*–P$_R$–*lux*)[53]. The lambda reporter is SOS responsive but not responsive to a TF–smORF module. The goal was to determine whether any condition or compound was a general phage inducer.

### smRNA-FISH

Custom Stellaris RNA FISH probes were designed against sets of early lytic genes in each phage under study using the Stellaris RNA FISH Probe Designer v.4.2 (LGC, Biosearch Technologies; Supplementary Table 7). Probes were labelled with Quasar 670 (phage 63 and phage ARM81ld) or CAL Fluor Red 590 (phage 72 and phage ARM81mr) (Biosearch Technologies). smFISH was performed as previously described[54], with minor modifications. In brief, cells were fixed with 1 ml cold 1× PBS and 3.7% formaldehyde and mixed at room temperature for 30 min. Samples were subjected to centrifugation at 400*g* for 8 min and the clarified supernatant discarded. Cells in the pellets were washed twice with 1× PBS and subjected to centrifugation at 600*g* for 3.5 min following each wash. Cells in the pellets were resuspended in 300 µl water and permeabilized by the addition of 700 µl 100% ethanol with mixing at room temperature for 1 h. Samples were subjected to centrifugation

at 600*g* for 7 min and the supernatant discarded. Cells in the pellets were resuspended in 1 ml Stellaris RNA FISH wash buffer A (Biosearch Technologies) containing 10% formamide, mixed for 5 min at room temperature, and the cells were collected by centrifugation at 600*g* for 7 min. Cells in the pellets were resuspended in 50 µl Stellaris RNA FISH hybridization buffer containing 10% formamide and 4 µl of each probe stock (12.5 µM) and incubated overnight at 37 °C. A 10 µl aliquot of the hybridization reaction was added to 200 µl wash buffer A followed by centrifugation at 600*g* for 3.5 min. Samples were resuspended in 200 µl wash buffer A, incubated for 30 min at 37 °C and subjected to centrifugation at 600*g* for 3.5 min. This step was repeated. Cells were stained with 50 µg ml$^{-1}$ DAPI in wash buffer A for 20 min at 37 °C, washed with 200 µl Stellaris RNA FISH wash buffer B and resuspended in 10 µl 1× PBS. Next, 1 µl aliquots of these cell samples were dispensed into No. 1.5 glass coverslip bottomed 24-well plates (MatTek), along with 30 µl VectaShield mounting medium and the samples were covered with agarose pads. Imaging was performed with a Nikon Eclipse Ti2 inverted microscope equipped with a Yokogawa CSU-W1 SoRa confocal scanning unit. Samples were imaged with a CFI Apochromat TIRF ×60 oil objective lens (Nikon, 1.49 numerical aperture) with excitation wavelengths of 405, 561 and 640 nm and with 0.4 µm (for *Aeromonas* phages) or 1 µm (for *Vibrio* phages) *z*-steps. Images were captured through a ×2.8 SoRa magnifier. Images were processed using Nikon NIS-Elements Denoise.ai software.

Cells in the images were segmented in Fiji software from maximum *z*-projections of the DAPI channel. Groups of cells that could not be accurately resolved were excluded from downstream analysis and coordinates of remaining cells were exported. smFISH data were analysed using custom Python scripts. In brief, images were convolved with a Gaussian function to remove noise. Spots were next detected as local maxima with intensities greater than a threshold set based on a negative control image (negative controls for each probe set are as follows: phage 72, *smORF*$_{63}$ induced; phage 63, *smORF*$_{72}$ induced; phage ARM81mr, *smORF*$_{ARM81ld}$ induced; phage ARM81ld, no induction). Each spot was fitted with a 3D Gaussian function to determine the integrated, background-subtracted spot intensity. In instances with multiple spots residing in close proximity, a 3D multi-Gaussian fit was performed. Spots were assigned to cells and the summed intensity from all spots in a cell were reported. Cells with total intensities ≤1 were assigned a pseudovalue of 1. Supplementary Table 8 provides summary data for *Vibrio* 1F-97 phage 72 and phage 63 and Supplementary Table 9 provides summary data for *Aeromonas* sp. phage ARM81ld and phage ARM81mr.

## In vitro HALO-cI repressor cleavage and in-gel HALO detection
Assessment of cleavage of HALO-cI proteins in response to DNA damage or smORF induction was carried out in *E. coli* according to a previously described method[19], with minor modifications. In brief, overnight cultures of *E. coli* T7Express lysY/I$^q$ carrying the indicated HALO fusion plasmid and cognate, aTc-inducible smORF vector were diluted 1:200 in medium and grown for 2.5 h with shaking. IPTG (200 µM) was added to the cultures before they were divided into 3 equal volumes followed by administration of the relevant treatment as specified. The treated cultures were incubated without shaking for an additional 2.5 h. Cells were collected by centrifugation (16,100*g* for 1 min), resuspended in BugBuster containing 1 µM HALO-Alexa$_{660}$ (excitation/emission of 663/690 nm). The cleared supernatant, collected after centrifugation of the lysate (16,100*g* for 10 min), was loaded onto a 4–20% SDS–PAGE stain-free gel. Gels were imaged using an ImageQuant LAS 4000 imager under the Cy5 setting for HALO-Alexa$_{660}$. Exposure times never exceeded 30 s.

## Protein production and purification
Plasmids harbouring genes encoding HIS–HALO–cI$_{Apop}$ and HIS–HALO–cI$_{ARM81ld}$ were introduced into *E. coli* BLR21(DE3) (Millipore Sigma) and a plasmid carrying HIS–HALO–cI$_{63}$ was introduced into

*E. coli* BL21(DE3) (Invitrogen). For protein production, the strains were grown overnight at 15 °C with 1 mM IPTG. In all cases, cells were pelleted at 3,000*g* for 10 min followed by resuspension in buffer A (150 mM NaCl, 20 mM Tris pH 7.5, 1 mM TCEP). Complete EDTA-free protease inhibitor cocktail tablets (Millipore Sigma) and 10 units of DNase I (Thermo) were added to each resuspended pellet and the cells were lysed by sonication. The insoluble fractions were separated from the soluble material by centrifugation of the lysates at 26,000*g* for 40 min. The soluble fractions were collected and applied to Ni-NTA Superflow resin (Qiagen). The resin was washed with 5 column volumes (CVs) of buffer A. Proteins were eluted in 2.5 CVs of buffer A and a gradient of 100–300 mM imidazole. The HIS–HALO tags were cleaved from the cI$_{Apop}$, cI$_{ARM81ld}$ and cI$_{63}$ proteins by treatment with 1 mg HIS-TEV Plus protease overnight at 4 °C. The samples were re-applied to Ni-NTA Superflow resin and the proteins were captured at around 85–95% purity by washing the resin with 1 CV of buffer A.

cI$_{72}$–HALO–HIS protein was produced in *E. coli* BLR21(DE3) by growth at 37 °C for 3 h with 1 mM IPTG. cI$_{72}$–HALO–HIS protein was purified from cell pellets as described above for the other cI proteins; however, to circumvent insolubility and aggregation, the HALO–HIS tag was not cleaved. Following initial purification on Ni-NTA Superflow resin, the eluate was concentrated and loaded onto a Superdex-200 size exclusion column (GE Healthcare) in Buffer A. cI$_{72}$–HALO–HIS at about 85% purity was collected.

## EMSA
DNA probes P$_{R63}$, P$_{R72}$, P$_{R-Apop}$ and P$_{R-ARM81ld}$ were generated using plasmids JSS-3131, JSS-3129, JSS-3346k and JSS-3348k, respectively, and the primers listed in Supplementary Table 5. Each EMSA reaction contained 20 ng of DNA probe (8–14 nM depending on the probe). The concentration of cI$_{Apop}$ and cI$_{ARM81ld}$ protein used in each reaction ranged from 200 nM to 800 nM. Higher concentrations of the cI$_{63}$ and cI$_{72}$–HALO–HIS proteins (800 nM to 3,200 nM) were required owing to their limited solubility. The cI proteins with their respective probes were combined in binding buffer (50 mM NaCl, 20 mM Tris pH 7.5, 1 mM TCEP) and incubated at room temperature for 15 min. The samples were subjected to electrophoresis on a Novex 6% DNA retardation gel (Thermo) in 1× TBE at 100 V for 45 min. Double-stranded DNA was stained with SYBR Green I nucleic acid gel stain (Thermo) for 20 min. After washing with 1× TBE, gels were imaged using an ImageQuant LAS 4000 imager under the SYBR Green setting.

## AlphaFold and DALI structural prediction of phage TF$_{72}$ and TF$_{63}$
AlphaFold2 (ref. 32) and AlphaFold-Multimer[55] were used to predict the structures of TF$_{72}$ and TF$_{63}$ from the protein sequences. The predicted structures were uploaded to the DALI server[56] for a heuristic PDB search. Structural predictions and homologues predicted by the DALI server were aligned and visualized using PyMOL[57].

## Quantitation and statistical analyses
The following software were used to collect and analyse data generated in this study: GraphPad Prism 9 for analyses of growth and reporter-based experiments; Gen5 v.3.11 for collection of growth and reporter-based data; MetaGeneMark v.3.26, HMMER3 v.3.3.2, VIBRANT v.1.2, MUSCLE v.3.8.31, Geneious Prime v.2022.2.2, SnapGene v.6, Prodigal 2.6.3, PhyML v.3.3.20180621, CD-HIT v.4.8.1 and BLAST 2.13.0+ for analyses of publicly available data and primer design; QuantStudio for qPCR data collection; Nikon NIS-Elements Denoise.ai for acquisition of FISH micrographs; and FIJI v.2.9.0/1.53r and Python v.3.7.6 for image analyses. AlphaFold 2.1.1, PyMOL Open GL 2.1 and DALI (accessed 26 November 2022) were used for protein structural predictions and analyses. Data are presented as the mean ± s.d. with *n* = 3 biological replicates starting from separate bacterial colonies measured on the same day. The number of technical and independent biological replicates for each experiment are indicated in the figure legends.

## Reporting summary

Further information on research design is available in the Nature Portfolio Reporting Summary linked to this article.

## Data availability

Unprocessed gels and micrographs are provided in Supplementary Fig. 1. All materials associated with this study are also deposited on Zenodo (https://doi.org/10.5281/zenodo.7083051). The accession codes for proteins presented in Extended Data Fig. 5b are provided in the corresponding legend and can be publicly accessed at Protein Data Bank with the following identifiers: 1Y7Y, 2B5A, 3G5G and 5WOQ. Other experimental data that support the findings of this study are available without restriction by request from the corresponding author. Source data are provided with this paper.

## Code availability

Custom code used to search, extract and analyse phage genome databases based on user-defined features and custom code for smFISH analysis are deposited on Zenodo (https://doi.org/10.5281/zenodo.7083051).

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

**Acknowledgements** We thank M. Polz (University of Vienna) for gifting us *Vibrio* 1F-97 and for thoughtful discussions throughout; M. Seyedsayamdost for providing the compound screening library; C. Fei for assistance with smRNA-FISH analyses; and J. Chen (Broad Institute), A. Biswas (Princeton University) and all members of the Bassler Laboratory for insightful discussions. This work was supported by the Howard Hughes Medical Institute, National Science Foundation grant MCB-1713731 and the National Institutes of Health grant R37GM065859 to B.L.B. J.E.S. is a Howard Hughes Medical Institute Fellow of the Jane Coffin Childs Memorial Fund for Medical Research. O.P.D. was supported by the NIGMS T32GM007388 grant. G.E.J. is a Howard Hughes Medical Institute Fellow of the Damon Runyon Cancer Research Foundation, DRG-2468-22. G.A.B. was supported by NIH Grant F32GM149034. F.A.H. is funded by the Schmidt Science Fellowship. K.J.F. was supported by the Endowed Scholars Program at the University of Texas Southwestern Medical Center, a Searle Scholars award and NIH grant 1DP2-AI154402. The content is solely the responsibility of the authors and does not necessarily represent the official views of the funders.

**Author contributions** J.E.S., O.P.D., F.A.H., K.J.F. and B.L.B. conceptualized the project and designed experiments. J.E.S. and O.P.D. constructed strains. J.E.S. and O.P.D. performed growth, reporter, in vitro cleavage, small-molecule screening and qPCR assays. J.E.S., F.A.H. and K.J.F. performed bioinformatics analyses. G.E.J. performed smFISH experiments. G.A.B. performed protein prediction and EMSA experiments. All authors analysed data. J.E.S., O.P.D., G.E.J., G.A.B., K.J.F. and B.L.B. wrote the paper.

**Competing interests** The authors declare no competing interests.

### Additional information
**Correspondence and requests for materials** should be addressed to Bonnie L. Bassler.

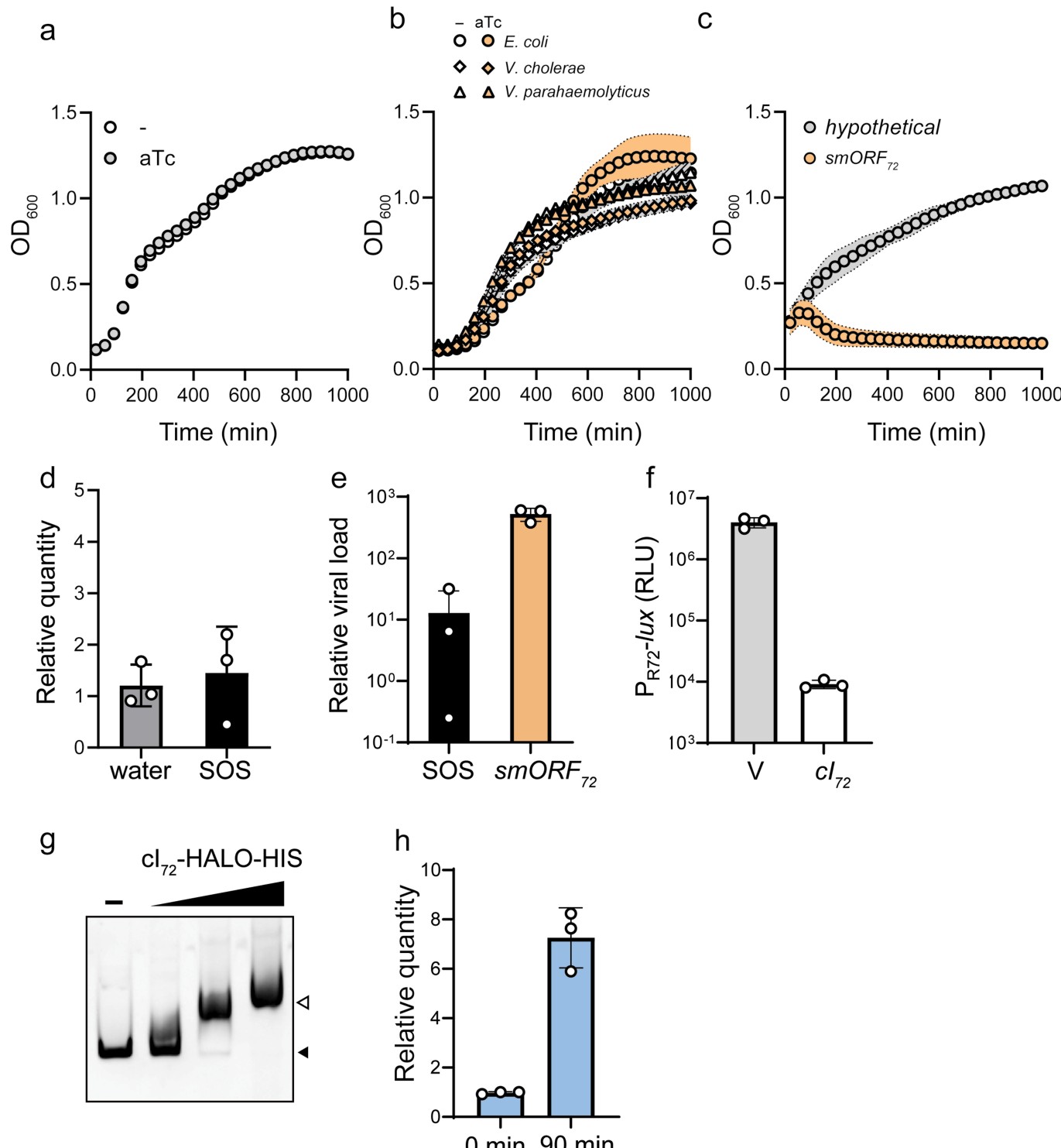

**Extended Data Fig. 1 | smORF$_{72}$-induced lysis of *Vibrio* 1F-97 is phage dependent and requires a TF-smORF module. (a)** Growth of WT *Vibrio* 1F-97 in medium lacking or containing aTc (white and gray, respectively). **(b)** Growth of *E. coli* (circles), *V. cholerae* (diamonds), and *V. parahaemolyticus* (triangles) carrying aTc-inducible *smORF$_{72}$* in medium lacking or containing aTc (white and orange, respectively). **(c)** Growth of WT *Vibrio* 1F-97 carrying an aTc-inducible *smORF* gene (designated *hypothetical*) that resides between *repA-telN* on phage 63 or aTc-inducible *smORF$_{72}$* in medium containing aTc (gray and orange, respectively). **(d)** Relative expression of *smORF$_{72}$* by RT-qPCR in *Vibrio* 1F-97 15 min after addition of water or ciprofloxacin. Relative transcript levels are the amount of phage *smORF$_{72}$* versus *rpoB* RNA, normalized to T = 0 min. **(e)** Detection of phage 72-specific particles in culture fluids from *Vibrio* 1F-97 carrying aTc-inducible *smORF$_{72}$* induced with ciprofloxacin or aTc. Relative viral load is the amount of *cI$_{72}$* DNA in induced versus uninduced samples by qPCR. **(f)** Expression of plasmid-borne P$_{R72}$-*lux* in *E. coli* containing an empty vector (V) or the phage *cI$_{72}$* gene. **(g)** EMSA showing binding of cI$_{72}$-HALO-HIS protein to P$_{R72}$ DNA. Approximately 14 nM of P$_{R72}$ DNA was combined with 800, 1600, or 3200 nM of cI$_{72}$-HALO-HIS protein. The no protein control is designated with a minus sign. Locations of the unshifted and shifted probe are indicated with black and white arrows, respectively. **(h)** Relative expression of *smORF$_{72}$* by RT-qPCR in *Vibrio* 1F-97 carrying aTc-inducible TF$_{72}$ at 0 and 90 min after induction with aTc. Relative transcript levels are the amount of phage *smORF$_{72}$* versus *rpoB* RNA, normalized to T = 0 min. Data are represented as means ± std with *n* = 3 biological replicates (**a,b,c,e,f**), as a single representative image from three independent experiments (**g**), and as means ± std with *n* = 3 biological replicates and *n* = 4 technical replicates (**d,h**). RLU as in Fig. 1d (**f**).

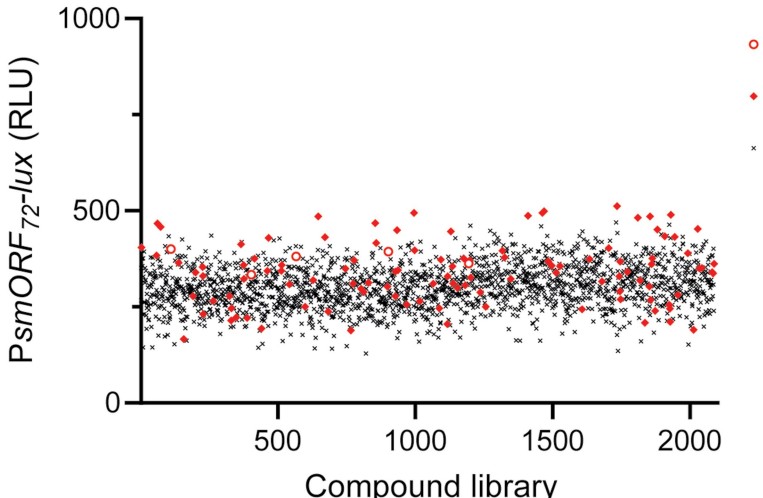

a

PsmORF$_{72}$-lux (RLU)

1000

500

500    1000    1500    2000

Compound library

○ water
◆ DMSO
× Compound

b

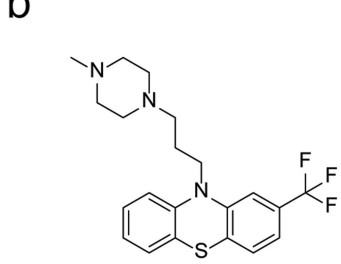
trifluoperazine

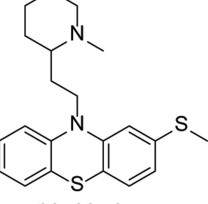
thioridazine

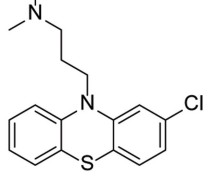
chlorpromazine

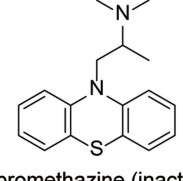
promethazine (inactive)

**Extended Data Fig. 2 | No *tf$_{72}$-smORF$_{72}$*-specific inducer was identified in two small-molecule screens. (a)** Relative P*smORF$_{72}$-lux* expression in *Vibrio* 1F-97 cultured in Biolog microarray plates and with a curated library of antibiotics. The red circles and diamonds show the water and DMSO vehicle controls, respectively. The black x symbols show the results for the different compounds tested. Fig. 1f shows assessment of reporter function. Conditions that did not support growth were excluded from analysis. Data are represented as a single reading. RLU as in Fig. 1d. **(b)** Structures of phenothiazines identified from the compound library that are not known DNA-damaging agents and that induce a phage lambda-derived *cI*-P$_R$-*lux* reporter (see Methods).

## a

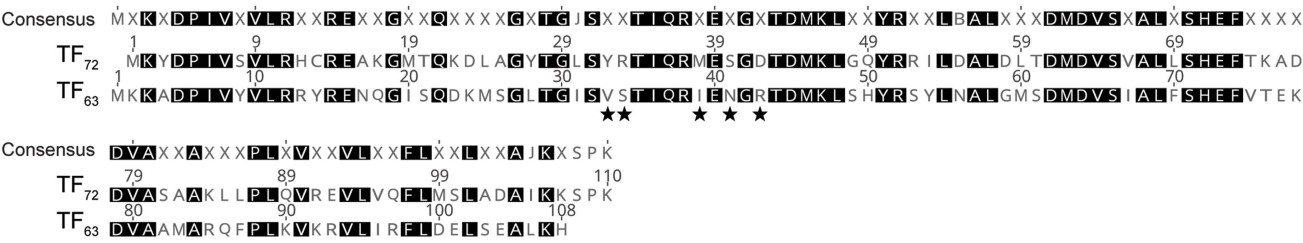

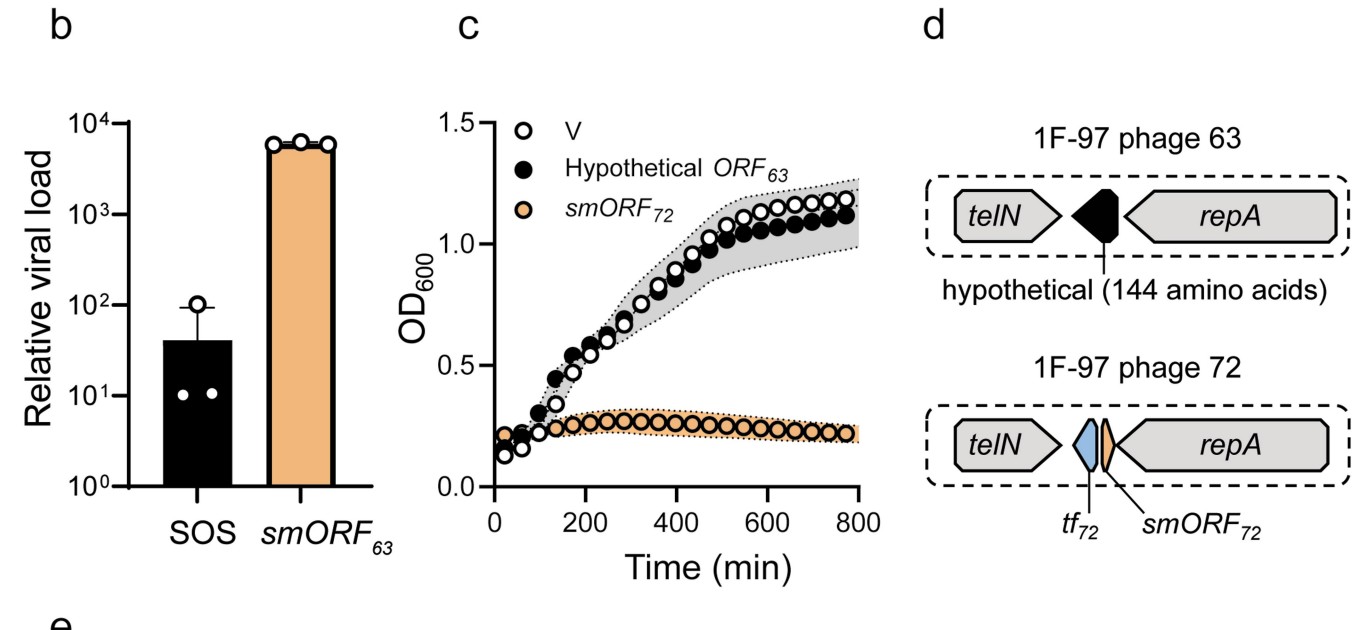

## b

## c

## d

## e

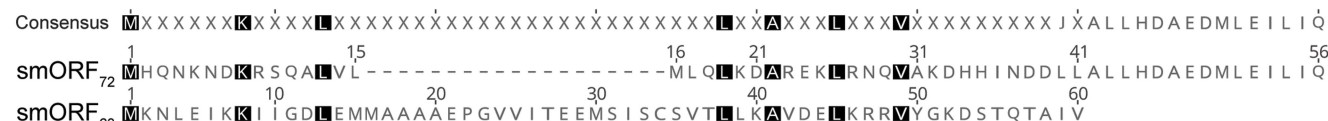

**Extended Data Fig. 3 | Sequence alignment of phage 72 and phage 63 TF proteins and characterization of the phage 63 TF$_{63}$-smORF$_{63}$ module in *Vibrio* 1F-97. (a)** Protein sequence alignment (ClustalW) showing TF$_{72}$ and TF$_{63}$. Black boxes show identical residues. The X symbols in the consensus sequence designate different residues. Stars below residues indicate the 5 amino acids in each protein that confer promoter specificity and that have been exchanged in TF$_{72-5x}$ and TF$_{63-5x}$ (see Fig. 2b). **(b)** Detection of phage 63-specific particles in viral preparations of culture fluids from *Vibrio* 1F-97 carrying aTc-inducible *smORF$_{63}$* that were grown in medium with ciprofloxacin or aTc to induce SOS and smORF$_{63}$ production, respectively. Relative viral load is the amount of *cI$_{63}$*

in the induced samples relative to an uninduced sample as judged by qPCR. **(c)** Growth of *Vibrio* 1F-97 carrying a plasmid containing the intervening gene between *repA* and *telN* from phage 63 under an aTc-inducible promoter (black), a plasmid carrying aTc-inducible *smORF$_{72}$* (orange), or no plasmid (white). All media contained aTc. **(d)** Organization of genes encoded between *repA* and *telN* in phage 63 and phage 72. **(e)** Protein sequence alignment as in (**a**), for smORF$_{72}$ and smORF$_{63}$. Colors and symbols as in (**a**). Data are represented as means ± std with *n* = 3 biological replicates (**c**) and as means ± std with *n* = 3 biological replicates and *n* = 4 technical replicates (**b**).

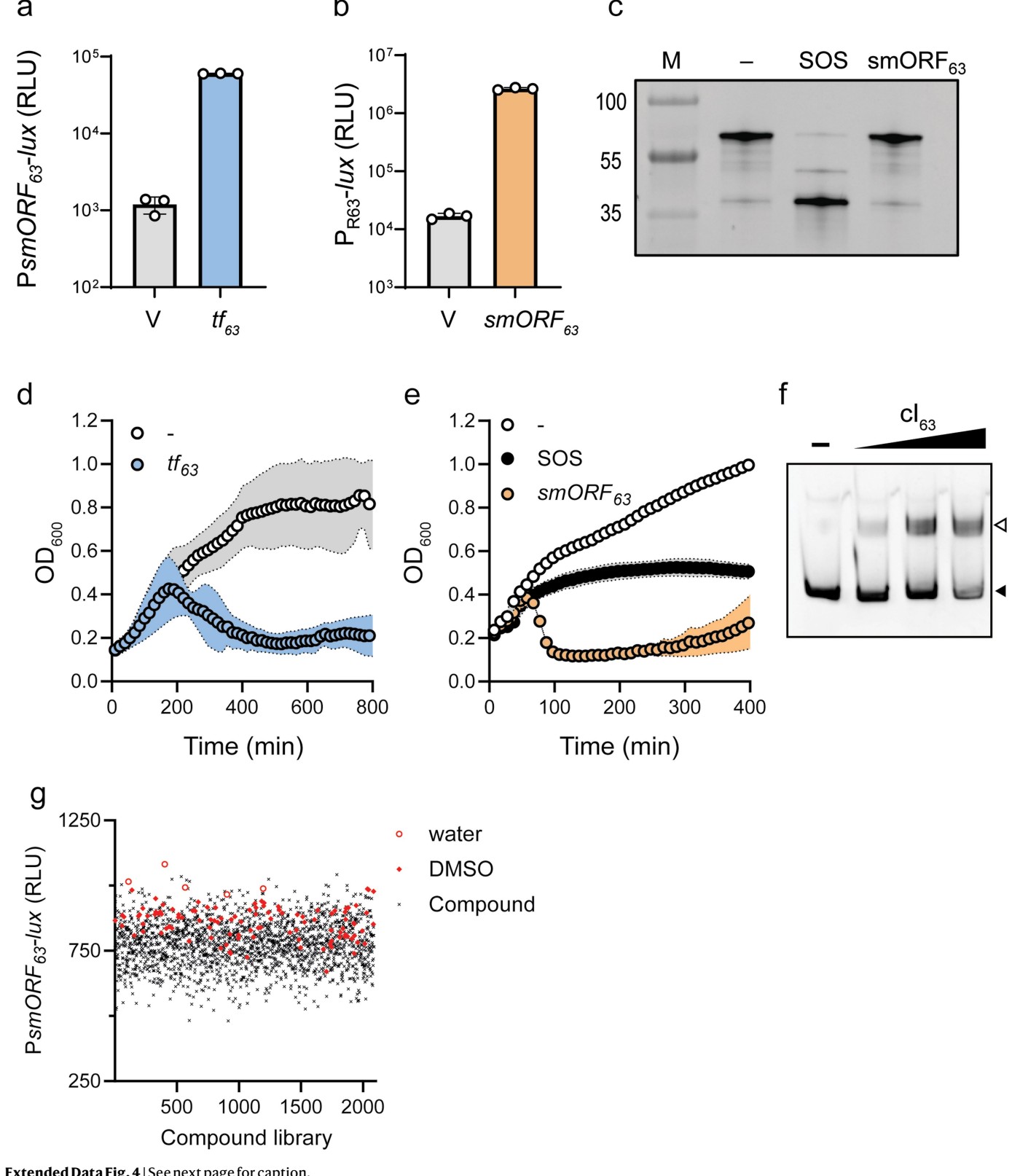

**Extended Data Fig. 4** | See next page for caption.

**Extended Data Fig. 4 | TF$_{63}$ activates transcription of *smORF$_{63}$* and smORF$_{63}$ non-proteolytically inhibits cI$_{63}$, driving host-cell lysis. (a)** P*smORF$_{63}$*-*lux* expression from *E. coli* carrying an empty vector (V) or aTc-inducible *tf$_{63}$* in medium containing aTc. **(b)** P$_{R63}$-*lux* expression in *E. coli* carrying an empty vector (V) or aTc-inducible *smORF$_{63}$* in medium containing aTc. The P$_{R63}$-*lux* plasmid carries *cI$_{63}$*, which natively represses reporter expression. **(c)** SDS-PAGE in-gel labeling of the phage 63 repressor (HALO-cI$_{63}$) produced in *E. coli* carrying aTc-inducible *smORF$_{63}$*. The treatments -, SOS, and smORF$_{63}$ refer to water, ciprofloxacin, and aTc, respectively. M as in Fig. 1b. **(d)** Growth of *Vibrio* 1F-97 carrying aTc-inducible *tf$_{63}$* in medium lacking or containing aTc (white and blue, respectively). **(e)** Growth of *Vibrio* 1F-97 carrying aTc-inducible *smORF$_{63}$* in medium containing aTc (designated *smORF$_{63}$*, orange), ciprofloxacin (designed SOS, black), or water (designated -, white). **(f)** EMSA showing binding of cI$_{63}$ protein to P$_{R63}$ DNA. Approximately 8 nM of P$_{R63}$ DNA was combined with 800, 1600, or 3200 nM of cI$_{63}$ protein. The no protein control lane is designated with a minus sign. Locations of the unshifted and shifted probe are indicated with black and white arrows, respectively. **(g)** Relative P*smORF$_{63}$*-*lux* expression in *Vibrio* 1F-97 cultured in Biolog microarray plates and with a curated library of antibiotics. The red circles and diamonds represent water and DMSO vehicle controls, respectively. The black x symbols show the results for the different compounds tested. Assessment of reporter function is provided in (**a**). RLU as in Fig. 1d (**a**,**b**,**g**). Data are represented as means ± std with $n = 3$ biological replicates (**a**,**b**,**d**,**e**), as a single reading (**g**), and as a single representative image from three independent experiments (**f**).

**Extended Data Fig. 5 | Phage TFs are predicted to be structurally similar to Restriction-Modification system controller proteins and to ClgR.** (a) AlphaFold predictions for $TF_{72}$ (brown) and $TF_{63}$ (blue) shown as superimposed homodimers. Secondary structural alpha helix elements are labeled. N and C termini are labeled N, N' and C, and C', respectively. (b) Structural alignment of $TF_{72}$ (brown) and $TF_{63}$ (blue) as monomers with the highest scoring homologs: C.AhdI (green), C.BclI (orange), C.Esp1396I (pink), and ClgR (gray) (PDB ID: 1Y7Y, 2B5A, 3G5G, and 5WOQ, respectively).

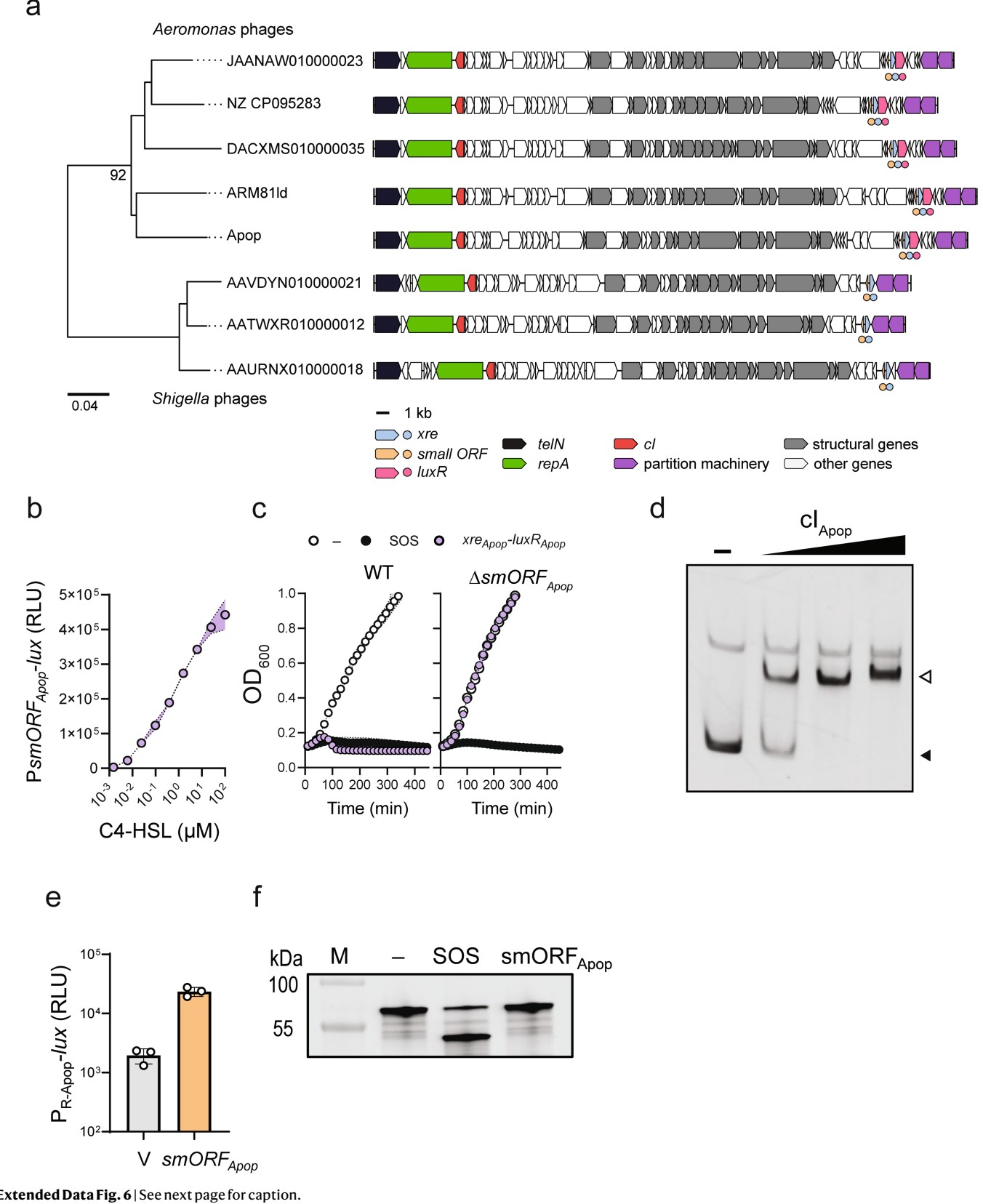

**Extended Data Fig. 6** | See next page for caption.

**Extended Data Fig. 6 | Comparison of linear plasmid-like phages in *Shigella* and *Aeromonas* reveals similar locations of genes encoding TF-smORF regulatory modules and HSL-quorum-sensing-receptor TF-smORF modules that control lysis. (a)** Phylogenetic tree (left) of 5 *Aeromonas* phages encoding *xre-luxR* genes and 3 *Shigella* phages encoding *xre*. The genome organization for each phage is depicted at the right. Genes are colored by annotation as noted in the key. Circles denote the locations of relevant features that are common among the 8 phages (*xre* (blue) and *smORF* (orange)) or exclusive to the 5 *Aeromonas* phages (*luxR* (pink)). *Shigella* phages are labeled with their corresponding NCBI accession numbers. Numbers above branches are pseudo-bootstrap support values from 100 replications. **(b)** P*smORF*$_{Apop}$-*lux* expression from *E. coli* carrying aTc-inducible *xre*$_{Apop}$-*luxR*$_{Apop}$ in medium with aTc and the indicated concentrations of C4-HSL. **(c)** Growth of *A. popoffii* carrying WT Apop (left) or Δ*smORF*$_{Apop}$ Apop (right), each harboring aTc-inducible *xre*$_{Apop}$-*luxR*$_{Apop}$

and grown in medium containing 5 ng mL$^{-1}$ aTc (purple), 1 μg mL$^{-1}$ ciprofloxacin (black), or water (white). All media contained 10 μM C4-HSL. **(d)** EMSA showing binding of cI$_{Apop}$ protein to P$_{R-Apop}$ DNA. Approximately 10 nM of P$_{R-Apop}$ DNA was combined with 200, 400, or 800 nM of cI$_{Apop}$ protein. Locations of the unshifted and shifted probe are indicated with black and white arrows, respectively. **(e)** P$_{R-Apop}$-*lux* expression from *E. coli* carrying an empty vector (V) or aTc-inducible *smORF*$_{Apop}$ in medium containing aTc. The P$_{R-Apop}$-*lux* plasmid carries two copies of *cI*$_{Apop}$ (see Methods) for native repression of reporter expression. **(f)** SDS-PAGE in-gel labeling of the Apop repressor (HALO-cI$_{Apop}$) produced in *E. coli* carrying aTc-inducible *smORF*$_{Apop}$. The treatments -, SOS, and smORF$_{Apop}$ refer to water, ciprofloxacin, and aTc, respectively. M as in Fig. 1b. Data are represented as means ± std with *n* = 3 biological replicates (**b,c,e**), and as a single representative image from three independent experiments (**d,f**) RLU as in Fig. 1d (**b,e**).

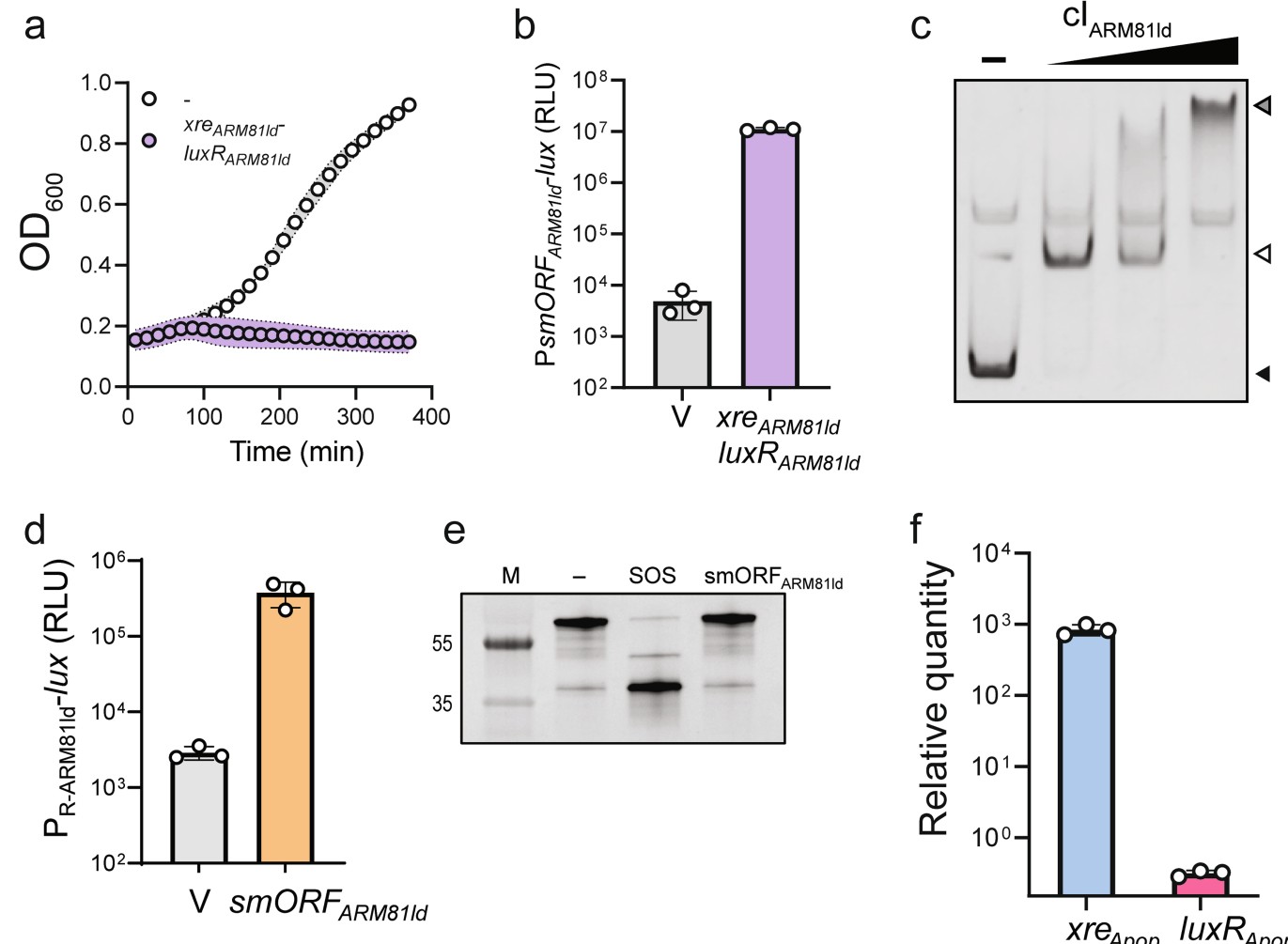

**Extended Data Fig. 7 | The TF-smORF modules of the Apop and ARM81ld phages function analogously, and both XRE and LuxR are required to bind and activate the cognate P*smORF*. (a)** Growth of *Aeromonas* sp. ARM81 harboring aTc-inducible *xre*$_{ARM81ld}$-*luxR*$_{ARM81ld}$ in the presence and absence of aTc (purple and white, respectively). **(b)** P*smORF*$_{ARM81ld}$-*lux* expression in *E. coli* carrying an empty vector (V) or aTc-inducible *xre*$_{ARM81ld}$-*luxR*$_{ARM81ld}$ in medium containing aTc and C4-HSL. **(c)** EMSA showing binding of cI$_{ARM81ld}$ protein to P$_{R\text{-}ARM81ld}$ DNA. Approximately 10 nM of P$_{R\text{-}ARM81ld}$ DNA was combined with 200, 400, or 800 nM of cI$_{ARM81ld}$ protein. The no protein control lane is designated with a minus sign. Locations of the unshifted, shifted, and supershifted probe are indicated with black, white, and gray arrows, respectively. **(d)** P$_{R\text{-}ARM81ld}$-*lux* expression from *E. coli* carrying an empty vector (V) or aTc-inducible *smORF*$_{ARM81ld}$ in medium containing aTc. The P$_{R\text{-}ARM81ld}$-*lux* plasmid carries two copies of *cI*$_{ARM81ld}$ (see Methods) for native repression of reporter expression. **(e)** SDS-PAGE in-gel labeling of the ARM81ld repressor (HALO-cI$_{ARM81ld}$) produced in *E. coli* carrying aTc-inducible *smORF*$_{ARM81ld}$. The treatments -, SOS, and smORF$_{ARM81ld}$ refer to water, ciprofloxacin, and aTc, respectively. M as in Fig. 1b. **(f)** Relative transcript levels of the *xre*$_{Apop}$-*luxR*$_{Apop}$ operon from the Apop genome following plasmid expression of either aTc-inducible *xre*$_{Apop}$ or *luxR*$_{Apop}$ in *A. popoffii*. All media contained C4-HSL and aTc. Primer pairs specific to the intergenic region in the *xre*$_{Apop}$-*luxR*$_{Apop}$ locus but absent from the aTc-inducible *xre*$_{Apop}$ and *luxR*$_{Apop}$ plasmids were used to measure native *xre*$_{Apop}$-*luxR*$_{Apop}$ expression (see Methods). Relative transcript levels are the amount of *xre*$_{Apop}$-*luxR*$_{Apop}$ DNA relative to the amount of *rpoB* DNA, normalized to the sample overexpressing *luxR*$_{Apop}$. Data are represented as means ± std with *n* = 3 biological replicates (**a**,**b**,**d**), as means ± std with *n* = 3 biological replicates and *n* = 4 technical replicates (**f**), and as a single representative image from three independent experiments (**c**,**e**). RLU as in Fig. 1d (**b**,**d**).

# Reporting Summary

## Statistics

For all statistical analyses, confirm that the following items are present in the figure legend, table legend, main text, or Methods section.

| n/a | Confirmed | |
|---|---|---|
| ☐ | ☒ | The exact sample size (*n*) for each experimental group/condition, given as a discrete number and unit of measurement |
| ☐ | ☒ | A statement on whether measurements were taken from distinct samples or whether the same sample was measured repeatedly |
| ☒ | ☐ | The statistical test(s) used AND whether they are one- or two-sided<br>*Only common tests should be described solely by name; describe more complex techniques in the Methods section.* |
| ☒ | ☐ | A description of all covariates tested |
| ☒ | ☐ | A description of any assumptions or corrections, such as tests of normality and adjustment for multiple comparisons |
| ☐ | ☒ | A full description of the statistical parameters including central tendency (e.g. means) or other basic estimates (e.g. regression coefficient) AND variation (e.g. standard deviation) or associated estimates of uncertainty (e.g. confidence intervals) |
| ☒ | ☐ | For null hypothesis testing, the test statistic (e.g. *F*, *t*, *r*) with confidence intervals, effect sizes, degrees of freedom and *P* value noted<br>*Give P values as exact values whenever suitable.* |
| ☒ | ☐ | For Bayesian analysis, information on the choice of priors and Markov chain Monte Carlo settings |
| ☒ | ☐ | For hierarchical and complex designs, identification of the appropriate level for tests and full reporting of outcomes |
| ☒ | ☐ | Estimates of effect sizes (e.g. Cohen's *d*, Pearson's *r*), indicating how they were calculated |

*Our web collection on statistics for biologists contains articles on many of the points above.*

## Software and code

Policy information about availability of computer code

| | |
|---|---|
| Data collection | Software used to collect data generated in this study consisted of Gen5 v3.11 for collection of growth and reporter-based data; MetaGeneMark v 3.26, HMMER3 v 3.3.2, VIBRANT v1.2, MUSCLE v3.8.31, Geneious Prime v 2022.2.2, SnapGene v6, Prodigal 2.6.3, PhyML v3.3.20180621, CD-HIT v4.8.1, and BLAST 2.13.0+ for analyses of publicly available data and primer design; QuantStudio for qPCR data collection; Nikon NIS-Elements Denoise.ai for acquisition of FISH micrographs; and AlphaFold 2.1.1, PyMOL Open GL 2.1, and DALI (accessed 11/26/2022) for protein structural predictions. |
| Data analysis | GraphPad Prism 9 was used for analysis of growth and reporter-based experiments. FIJI v2.9.0 / 1.53r and Python v3.7.6 was used for image analyses. Custom code used to search, extract, and analyze phage genome databases based on user-defined features, and custom code for smFISH analysis are freely available and are deposited on Zenodo (doi: 10.5281/zenodo.7083051). |

For manuscripts utilizing custom algorithms or software that are central to the research but not yet described in published literature, software must be made available to editors and reviewers. We strongly encourage code deposition in a community repository (e.g. GitHub). See the Nature Portfolio guidelines for submitting code & software for further information.

## Data

Policy information about availability of data

All manuscripts must include a data availability statement. This statement should provide the following information, where applicable:
- Accession codes, unique identifiers, or web links for publicly available datasets
- A description of any restrictions on data availability
- For clinical datasets or third party data, please ensure that the statement adheres to our policy

Data presented in all figure panels of this study are provided in Source Data Figure files and Source Data Extended Data Figure files. Unprocessed gels and micrographs are provided in Supplementary Figure 1. All materials associated with this study are also deposited on Zenodo (doi: 10.5281/zenodo.7083051). The accession codes for proteins presented in Extended Data Figure 5b are provided in the corresponding legend and can be publicly accessed via PDB ID: 1Y7Y, 2B5A, 3G5G, and 5WOQ. Other experimental data that support the findings of this study are available without restriction by request from the corresponding author.

## Research involving human participants, their data, or biological material

Policy information about studies with human participants or human data. See also policy information about sex, gender (identity/presentation), and sexual orientation and race, ethnicity and racism.

| | |
|---|---|
| Reporting on sex and gender | Not applicable |
| Reporting on race, ethnicity, or other socially relevant groupings | Not applicable |
| Population characteristics | Not applicable |
| Recruitment | Not applicable |
| Ethics oversight | Not applicable |

Note that full information on the approval of the study protocol must also be provided in the manuscript.

# Field-specific reporting

Please select the one below that is the best fit for your research. If you are not sure, read the appropriate sections before making your selection.

☒ Life sciences  ☐ Behavioural & social sciences  ☐ Ecological, evolutionary & environmental sciences

For a reference copy of the document with all sections, see nature.com/documents/nr-reporting-summary-flat.pdf

# Life sciences study design

All studies must disclose on these points even when the disclosure is negative.

| | |
|---|---|
| Sample size | Sample size was chosen as three biological replicates, matching the standard in the microbiology field [e.g., Erez and Steinberger-Levy et al. Nature 541, 488–493 (2017)]. All datapoints displayed in this study are available in the source data for others to access and analyze. Means and standard deviations are plotted; no additional statistical analyses were performed [see D.L. Vaux "Know when your numbers are significant. Nature 492, 180–181 (2012)]. |
| Data exclusions | No data were excluded. |
| Replication | At least three replicates were used for each experiment. All data points were plotted and are available in the source data file. No data were excluded, and all replicates were therefore considered "successful" measurements. |
| Randomization | Randomization was not formally implemented in this study, however, the choice of wells and positioning of culture tubes used in any given experiment was not pre-assigned and was therefore chosen randomly at the time of setup. |
| Blinding | Blinding was not formally applied in this study. The investigators setting up the assays also analyzed the data. The strains used in each experiment were assigned numbers that were cross-checked with the corresponding sample names/treatments only at the time the data were plotted. |

# Reporting for specific materials, systems and methods

We require information from authors about some types of materials, experimental systems and methods used in many studies. Here, indicate whether each material, system or method listed is relevant to your study. If you are not sure if a list item applies to your research, read the appropriate section before selecting a response.

## Materials & experimental systems

| n/a | Involved in the study |
|-----|----------------------|
| ☒ | ☐ Antibodies |
| ☒ | ☐ Eukaryotic cell lines |
| ☒ | ☐ Palaeontology and archaeology |
| ☒ | ☐ Animals and other organisms |
| ☒ | ☐ Clinical data |
| ☒ | ☐ Dual use research of concern |
| ☒ | ☐ Plants |

## Methods

| n/a | Involved in the study |
|-----|----------------------|
| ☒ | ☐ ChIP-seq |
| ☒ | ☐ Flow cytometry |
| ☒ | ☐ MRI-based neuroimaging |

