## [Peer Review File · Nature]

Manuscript Title: Small protein modules dictate prophage

fates during polylysogeny

Reviewer Comments & Author Rebuttals

Reviewer Reports on the Initial Version:

Referees' comments:

Referee #1 (Remarks to the Author):

In this paper Justin Silpe and colleagues provide new insights into the way some linear phages control lysis/lysogeny decisions. They identify regulators that are suggested to enable phage to respond to different cues in order to transition into lytic replication. Moreover, it is suggested these regulators are highly specific and enable phage to induce from polylysogens without triggering induction of other prophages - unlike SOS reposes that typically induce multiple phages simultaneously.

The subject area is very interesting, and there is no doubt the manuscript contains several very interesting observations. The paper builds on seminal work from the same group (Silpe et al Cell 2019), which provided groundbreaking new insight into the mechanisms and cues that feed into lysis/lysogeny regulation by temperate phage. However, this study did not nearly excite me as much as their previous work, and I have several concerns related to data interpretation that further dampen my enthusiasm for this paper.

Major Comments:

1) It is nice that the authors identify new transcription factors and regulators of lysis/lysogeny for this particular groups of phages. For example, they find that TF72 regulates smORF72, which in turn appears to regulate the cl-pR72 interaction (and therefore the lysis/lysogeny switch). Likewise, they find that phage 63 carries a similar regulatory module, located elsewhere in the phage genome, that controls the lysis/lysogeny switch in a similar way. However, it remains unclear what these regulatory modules responds to. An exception to this is the Apop system, which - like the VP882 system discovered previously by the same lab - responds to host AI molecules. This is nice, but less novel given the findings described in Silpe et al Cell 2019. For the other linear phages, it remains unclear what cues they respond to. They may be within-cell signals, such as SOS or other stress responses, or extracellular signals. Given that this remains unclear, the paper provides only part of the story - the most important bit, i.e. the cues that feed into the lysis/lysogeny decision process, remain unknown. In my view, this severely limits the conceptual novelty, and the insight into the ways in which these phages regulate their infection dynamics.

2) The last part of the manuscript, where the authors over-express either ORF63 or ORF72 to induce a single phage within a poly-lysogen, also raises concerns. The authors infer that the regulators enable phage to respond to distinct cues to induce independent of one another. This is a major point of the paper that is presented in the abstract and in Figure 5e. However, this cannot be inferred from the data, because it is unclear what cues ORF63 and ORF72 themselves respond to. They might, as pointed out above, be the same cues, or they might be different cues. Without that knowledge, it is impossible to say anything at all about whether or not these phage can or will induce independent of one another. The assay, where a regulator that is specific for one phage is overexpressed, is artificial and excludes the key factor that acts upstream of (and feeds into) this regulatory element.

Other comments:

1) line 143-147 - critical control to exclude possibility that smORF72 is simply toxic to 1F-97 is lacking: the right control is to do the same experiment in a phage 72 deletion background.

2) line 150 - the authors state that smORF72 regulation acts independent of the SOS response, but this has not been demonstrated; this should be tested by performing the same experiment in a SOS KO strain.

3) line 331-337 The interpretation / verbal model that is provided may well be true, but this simply cannot be inferred from the data that are provided in the manuscript. This requires standard biochemical approaches.

Minor comments:

1) line 66-95 - interesting, but surely this can be written in a much more concise way?

2) line 135 - the authors decide to focus on 1F-97. I expected to see the gene neighbourhood in Fig. 1a, but it doesn't seem to be included. Why not? Where does it sit in the phylogeny? (I only later saw it is indicated in Figure 2a, but nonetheless, I think it should also be shown in Fig. 1a).

3) the telN repA intergenic region of phage 72 contains another gene, besides smORF72 and TF72. What is it? What domains does it carry? What is the predicted role? Does deletion of this gene impact lysis/lysogeny of phage TF72?

3) line 137 - putative transcription factor - how was this determined?

4) line 138-140 what about amino acid level or structural similarities (alphafold) ?

5) line 152-162 - here the authors carry out a simple lux reporter assay to track expression of the

pR72 promoter, and show that cI72 expression decreases and smORF72 increases expression levels from the pR72 promoter. The authors need to phase their findings more carefully given there is no biochemical data provided to demonstrate e.g. that cI72 acts on the pR72 promoter (line 155), or that smORF72 acts directly to inactivate cI72 (line 162). None of this is directly demonstrated, and therefore I feel the authors are over-interpreting the data.

6) line 164-172 the authors use a lux reporter assay to show that TF72 expression increases smORF72 expression (28-fold) in *E. coli* and in 1F-97 (7-fold), where it induces lysis. Again, no biochemistry, so a bit more care in the way things are phrased is recommended.

7) No information is given about the similarity (nucleotide, amino acid, predicted AlphaFold structures) of the TF63/TF72 and smORF63/smORF72 genes. From the text, I have the impression they are extremely closely related. Can the authors provide some alignments/structure predictions to clarify this?

8) Throughout, the authors focus all their assays on prophage induction, and never mention or measure lysogenization. Why is this? Is there a reason to assume the regulators are / aren't involved in this?

Referee #2 (Remarks to the Author):

Major Comments

A combination of bioinformatics and molecular genetics are used to identify a transcription factor and small protein circuit controlling the lysis decision in multiple prophages. This circuit is compared to the classical SOS response induction pathway (i.e., RecA). This work is an extension of the quorum sensing (QS)-induction work and illuminates several key components.

The paper is predicated on the premise that the phage field ubiquitously believes that DNA damage is the primary induction signal. This is an old view and the phage ecology literature is replete with models and experiments showing that DNA damage is not the major inducing factor. The manuscript would have more impact if the emphasis was placed on the "phage-on-phage war".

Overall, this is an intriguing work that is potentially of broad interest.

All of the "we" are distracting. Right now, the paper reads like an undergraduate discussing a night out..."We wondered", "We could not", "We synthesized", "We..." makes it almost impossible to follow the narrative. Similarly, the figures need to be improved. Add legends to the growth curves. I am a knowledgeable reader (i.e., I know main proteins & background for this work) and I had a very

hard time understanding what you did.

I really can't tell whether the data points are true replicates (i.e., from individual colonies on different days) or pseudoreplicates (i.e., the same culture sampled 3 times)? I think it is the latter, which means everything was effectively done from 1 colony to one overnight...Also, the experiments need to be repeated at least 5 times to make them more statistically sound.

Minor Comments

How common is the small protein (smORF) in the Short Read Archive?

There are several papers exploring novel inducing agents. It would be interesting to see which pathway these other agents cause induction.

Referee #3 (Remarks to the Author):

Summary of the article:

These authors discover and characterize novel lysis regulatory modules in linear plasmid-like phages. Inspired by the arrangement of a previously discovered regulatory module (VqmAphage and qtip), they bioinformatically searched through sequence databases and discovered several instances of a divergently transcribed transcription factor and a small open reading frame (smORF). Through heterologous expression of the genes singly and in combination in *E. coli*, they find that the smORF non-proteolytically interferes with the *cl* repressor of the phage that carries it, leading to phage induction. Through confocal microscopy, they show that the two smORFs from the two phages in the *Vibrio* strain they use act on the *c*-repressors in different ways. Further bioinformatic searches reveal TF-smORF modules that include a *luxR* gene, which they show is responsive to host-produced quorum-sensing molecules. Finally, they show that in polylysogens which contain these smORF-activated modules, expression of the smORF on one phage leads to near-exclusive production of that virus, creating a fitness advantage for that virus in that inducing condition. These results will be of interest to a more specific audience of those interested in the mechanisms of phage-host relationships and their implied ecological consequences.

The manuscript is clear and enjoyable (although long) read. The most exciting part of the paper is that it demonstrates a novel, phage specific, DNA-damage-independent mode of viral induction from bacterial genomes. The data to support this finding are compelling. Unfortunately, the clarity, elegance and impact of the story seem to be lost in the writing that combines two to three papers into one long and circuitous one.

The title, introduction, and discussion fit with the story that is very nicely told through the paragraph starting line 233. Then a long detour begins where the authors introduce a totally new concept

about the sequestering of the repressor to the cell poles and use Halo-tag to identify the localization of smORF72. This detour seems out of place and continues into a third direction with the XRE-LuxR genes of other phage in other species. As someone who is not an expert in the genetic methods, I felt these detours are unnecessary and story really loses momentum. The specificity and phage induction in a polylysogen which is FINALLY stated in the paragraph starting 451. While I was reading it I felt such relief when the authors came back around to their message. I had almost given up. I suggest cutting out the detours and sticking to one story for those who haven't read all of the other work.

Other comments:

- I found the logic stated on lines 75 – 78 for focusing on phage with RecA-dependent, autoproteolytic cI repressors is not clear. I read this several times and do not follow this reasoning for excluding other phage. Isn't diversity being lost?
- In a related question, on line 86: the statement "indicating that our autoproteolytic set less redundantly sampled nature's diversity" is unclear. Do you mean that less than what? The authors must know that the databases do not sample viruses or microbes in an unbiased way. This tells us nothing about "Nature's diversity" and the concept of "less redundantly" is very confusing. Please restate.
- The frequency of other small-ORF candidates throughout phage genomes, independently of gene neighborhood, was not assessed. The logic for keeping the search limited to gene neighborhoods is not clear. If the average length of these phages is estimated at around 50kb (and at least <100kb), casting a 10kb net around a parB sequence encompasses 40% of that prophage prediction, which decreases the chance that parB presence/absence is coupled to these regulatory modules. Line 253 cites the use of 25kb as a cutoff for searching for modules near "conserved phage structural genes". Many c-repressors are more than within that cutoff, making its use a confusing filter. How many small ORF/transcription pairs can be found without requiring an association with parB or repA, or when the presence of a different gene (like an integrase) is required? In general, the discovery of a TF-smORF pair that was not coupled with the originally queried repA-telN proteins lends itself to a broader search, not an additional, equally limiting search. These filtration steps are used as a lead-in to the discovery of the already-discovered and ready-to-use Aeromonads, which seems convenient and diminishes the integrity of the posed question. Additionally, the emphasis on "small protein modules" as suggested by the title is not reflected in the way these searches were conducted.
- There is no discussion of the integrity of the sequences that are put forward as putative phage genomes, at least in the NCBI nt database. The additional step at least of phage prediction from sequences of interest could have cut down on some of the need for arbitrary distance cutoffs. The lack of reference to recognizable lysogeny genes (integrases, transposases, c-repressors) in a paper about putative induction modules is glaring, as searches for putative induction machinery implies successful latent infections must have taken place. This problem is implicitly overcome by synthetic generation of gene cassettes in some cases, but should be explicitly referenced.
- Line 374: The heading "Variations in host bacterial signal transduction cascades..." implies that some phages have a "choice" of what transcription factor to use, where different transcription factors encoded by the same phage may act independently on these regulatory modules. However, the bioinformatic results of the XREApop tBLASTn search were not clearly described. Were only five viruses subsequently found, as described in the Extended Data Fig 3a? The question posed on line

377 has already been disproven earlier in the paper, which show extensive phage TF-smORF modules without a LuxR component. The XRE-LuxR modules were happened upon ostensibly as one of many TF-smORF pairs, and their treatment as a unique category limits the potential directions of the sensory inputs these phages are using. Any cursory investigation into Shigella quorum-sensing pathways which would reveal homologues on Shigella phages would be a more satisfying reason to include this section of the paper. Without it, the conclusion on line 395 is mystifyingly obvious.

- Given that this work was done from a pool of “61 unique loci” and “56 unique contigs” and “three contigs, likely from linear plasmid-like phages” it seems unlikely that at this stage it is reasonable to say on line 520 that “these pathways may in fact be the primary determinants of phage lifestyle transitions”. This work seems to specifically interrogate linear plasmid-like phages and doesn’t attempt to assess phages writ broad. What is known about the frequency and abundance of this phage type in nature? A frank discussion about this would be useful given the specificity of the findings. Line 377 admits that the investigation has taken place in “particular sets of regulatory components encoded on different plasmid-like phages”, limiting the applicability of this work to phages in general.

- The final paragraph of the discussion is an incompletely described “just so story” about the relevance of this mechanism to the phage. This is at best speculative and should absolutely be removed as it only hurts the impact of the paper overall.

Minor comments:

- Figure 2a could have been easily used as well to illustrate the lack of identity using Mauve or Clinker (helpful tool – see Gilchrist and Chooi, 2020, Bioinformatics). How similar are repA, telN, and the partition machinery?
- Line 189: share little identity..(Mauve), is this supposed to be a figure? A reference? A figure would be nice.
- Line 410: Figure 2a is referenced to support amino acid identity, but does not contain any amino acid identity information.
- The presence of autoproteolytic repressors is an interesting filter to use, but was not repeated in the search for parB-associated loci and could have been useful.
- The strain that was previously studied (VP882) is included in the tree in Fig 1A, but the *Vibrio cyclitrophicus* 1F-97 strain that is a major subject of study in this paper does not seem to be included in the tree. By contrast, the other investigated phages are included in the tree when they’re referenced.
- Lines 511-519 (“Presumably, the native cues... niche in which they reside”) are confusing in the context of phage induction, as the first cyanophage example suggests induction would not be favored, and the second suggests a purely lytic relationship with a bacterial host, without any need for multiple, finely-tuned induction pathways. Neither of them are suggestive of the dynamic relationship that is proposed to govern these phages, nor do they suggest what the referenced cues might be. See above.
- Line 3 is incorrect in light of several recent works highlighting the importance of chronic viruses in several different taxa, and is not a strong beginning to a strong paper. See especially, among several others, Roux et al, 2019 in Nature Microbiology.

Author Rebuttals to Initial Comments:

Referees' comments:

Referee #1 (Remarks to the Author):

In this paper Justin Silpe and colleagues provide new insights into the way some linear phages control lysis/lysogeny decisions. They identify regulators that are suggested to enable phage to respond to different cues in order to transition into lytic replication. Moreover, it is suggested these regulators are highly specific and enable phage to induce from polylysogens without triggering induction of other prophages - unlike SOS reposes that typically induce multiple phages simultaneously.

The subject area is very interesting, and there is no doubt the manuscript contains several very interesting observations. The paper builds on seminal work from the same group (Silpe et al Cell 2019), which provided groundbreaking new insight into the mechanisms and cues that feed into lysis/lysogeny regulation by temperate phage. However, this study did not nearly excite me as much as their previous work, and I have several concerns related to data interpretation that further dampen my enthusiasm for this paper.

Major Comments:

1) It is nice that the authors identify new transcription factors and regulators of lysis/lysogeny for this particular groups of phages. For example, they find that TF72 regulates smORF72, which in turn appears to regulate the cl-pR72 interaction (and therefore the lysis/lysogeny switch). Likewise, they find that phage 63 carries a similar regulatory module, located elsewhere in the phage genome, that controls the lysis/lysogeny switch in a similar way. However, it remains unclear what these regulatory modules responds to. An exception to this is the Apop system, which - like the VP882 system discovered previously by the same lab - responds to host AI molecules. This is nice, but less novel given the findings described in Silpe et al Cell 2019. For the other linear phages, it remains unclear what cues they respond to. They may be within-cell signals, such as SOS or other stress responses, or extracellular signals. Given that this remains unclear, the paper provides only part of the story - the most important bit, i.e. the cues that feed into the lysis/lysogeny decision process, remain unknown. In my view, this severely limits the conceptual novelty, and the insight into the ways in which these phages regulate their infection dynamics.

We agree with the Reviewer that knowledge of the cues to which the TF-smORFs studied here respond is of utmost importance. Indeed, identifying the native cues controlling expression and activity of these modules is a primary focus of our ongoing research, but thus far, our efforts have been unsuccessful. We have screened ~2,500 diverse compounds but have not yet found an inducer. In the revised manuscript, we now include the negative data and the details of the screens in Extended Data Figures 2a and 4g.

2) The last part of the manuscript, where the authors over-express either ORF63 or ORF72 to

induce a single phage within a poly-lysogen, also raises concerns. The authors infer that the regulators enable phage to respond to distinct cues to induce independent of one another. This is a major point of the paper that is presented in the abstract and in Figure 5e. However, this cannot be inferred from the data, because it is unclear what cues ORF63 and ORF72 themselves respond to. They might, as pointed out above, be the same cues, or they might be different cues. Without that knowledge, it is impossible to say anything at all about whether or not these phage can or will induce independent of one another. The assay, where a regulator that is specific for one phage is overexpressed, is artificial and excludes the key factor that acts upstream of (and feeds into) this regulatory element.

We agree with the Reviewer that it is possible that the TF-smORF modules encoded by phage 72 and phage 63 are induced by the same cue. AlphaFold structural predictions of TF₇₂ and TF₆₃ revealed these TFs most closely resemble each other (RMSD = 1.3 Å), restriction modification controller proteins (RMSD = 0.5 – 0.9 Å), and the helix-turn-helix (HTH) transcription factor, CIGR from *Mycobacterium smegmatis* (RMSD = 1.0 – 2.5 Å). These points are now included in the Results and Discussion (lines 199-203, respectively) along with an accompanying Extended Data Figure showing the structural predictions (Extended Data Figure 5a and b). Regarding the cues controlling the activities of these TFs, restriction modification controller proteins are subject to regulation at the level of transcription but are not known to bind small molecule ligands. CIGR, which is not a phage-encoded protein in *Mycobacterium*, regulates RecA-independent genes involved in stress response/repair (Wang et al. 2011 JBC, 10.1074/jbc.M111.241802), a curious parallel to those from the phages we study here.

Our conclusion that the specific induction cue detected determines the outcome of inter-prophage competition stemmed from experiments demonstrating specificity in induction of each of the two phages in the *Vibrio* 1F-97 polylysogen and each of the two phages in the *Aeromonas* sp. ARM81 polylysogen. In the latter case, only the ARM81ld phage harbors a TF-smORF module, thus it can be induced independently of the co-residing ARM81mr phage. To bolster this aspect of the work and to expand the conceptual novelty of the findings, in the revised manuscript we include RNA FISH experiments to visualize, for the first time, co-induction of phage 72 and phage 63 following DNA damage and independent induction of each phage via smORF₇₂ or smORF₆₃ overproduction. We also performed and provided analogous experiments for the *Aeromonas* sp. ARM81 phages. Interestingly, in the case of DNA-damage, in which both phages respond, we find that discrete subsets of cells produce either one phage, the other phage, or both phages. Thus, single and dual phage production is possible from an individual host cell. The new RNA FISH experiments are included in lines 321-345 of the manuscript and the data are shown in Figure 4b and 4c.

Other comments:

1) line 143-147 - critical control to exclude possibility that smORF72 is simply toxic to 1F-97 is lacking: the right control is to do the same experiment in a phage 72 deletion background.

We agree with the Reviewer, however, we have been unsuccessful in our efforts to cure phage 72 or phage 63 from *Vibrio* 1F-97. To circumvent this issue, we overexpressed smORF₇₂ in other *Vibrios* such as *Vibrio cholerae* and *Vibrio parahaemolyticus* as well as in *Escherichia coli*, which do not harbor phage 72 and we find that smORF₇₂ is not toxic. We now include that control experiment in lines 119-122 and the data are presented as Extended Data Figure 1a and 1b.

2) line 150 - the authors state that smORF72 regulation acts independent of the SOS response, but this has not been demonstrated; this should be tested by performing the same experiment in a SOS KO strain.

Despite many attempts, we remain unable to engineer modifications to the chromosome of *Vibrio* 1F-97, a newly discovered undomesticated strain. Thus, we could not make the $\Delta recA$ deletion requested by the Reviewer. Because we do agree that the Reviewers raises an important point, we devised an alternative approach to test the involvement of the SOS response in activating *tf-smORF* expression in WT *Vibrio* 1F-97. Specifically, we administered ciprofloxacin and used RT-qPCR to measure transcription of *smORF₇₂* shortly after SOS induction. *smORF₇₂* expression did not significantly change. Thus, we exclude SOS as being the the native inducer of *smORF₇₂*. This experiment is described in lines 126-128 and the data are shown in Extended Data Figure 1d.

3) line 331-337 The interpretation / verbal model that is provided may well be true, but this simply cannot be inferred from the data that are provided in the manuscript. This requires standard biochemical approaches.

As detailed below in our responses to this Reviewer's points 5 and 6, in the revised manuscript, we have successfully expressed, purified, and demonstrated *in vitro* DNA-binding for each of the *cl* repressors under study. However, despite numerous trials, we could not overexpress the *tf* genes to a level that would enable purification of the TF proteins. Nonetheless, to provide additional insight into the DNA-binding specificity of each *Vibrio* 1F-97 phage TF, we use structural predictions to pinpoint the DNA-binding domains. We now show that exchange of only 5 amino acids in this region reverses these promoter specificities. The data are provided in Figure 2b and the text describing this experiment appears at lines 204-207.

As noted in our response to point 6 from this Reviewer, the *Aeromonas* phage systems, which are considerably more complex than those in *Vibrio* 1F-97 because they depend on multiple proteins (XRE and LuxR) and a ligand, await similar biochemical testing *in vitro*. Given the genetic evidence we provide in both *E. coli* and *Aeromonas*, we consider it reasonable to put forth our proposed model. In the manuscript, we are careful not to claim that XRE and LuxR form a protein-protein interaction, but we do offer it as one possibility in Figure 3e and the associated legend.

Minor comments:

1) line 66-95 - interesting, but surely this can be written in a much more concise way?

We agree with the Reviewer and have now removed lines 83-90 from the original manuscript to consolidate this section.

2) line 135 - the authors decide to focus on 1F-97. I expected to see the gene neighbourhood in Fig. 1a, but it doesn't seem to be included. Why not? Where does it sit in the phylogeny? (I only later saw it is indicated in Figure 2a, but nonetheless, I think it should also be shown in Fig. 1a).

We apologize for this omission and thank the Reviewer for highlighting our oversight. Phage 72 has always been present in this figure (accession KP795532), and is now explicitly indicated with a star.

3) the *telN* *repA* intergenic region of phage 72 contains another gene, besides *smORF72* and *TF72*. What is it? What domains does it carry? What is the predicted role? Does deletion of this gene impact lysis/lysogeny of phage *TF72*?

The additional predicted gene encoded between *repA*-*telN* is of unknown function, with no homologs or predicted domains. We overexpressed this gene in *Vibrio* 1F-97 and there was no measurable effect on growth in our assays. This experiment is now included in lines 123-125 and the data are shown in Extended Data Figure 1c.

3) line 137 - putative transcription factor - how was this determined?

We determined that *TF72* is a putative transcription factor via the NCBI conserved domain database. We now note that in line 111 of the text.

4) line 138-140 what about amino acid level or structural similarities (alphafold) ?

As requested, we have now provided additional figures showing the amino acid sequence alignments for *TF72* and *TF63* (Extended Data Figure 3a) and for *smORF72* and *smORF63* (Extended Data Figure 3e). As noted in the manuscript, *TF72* and *TF63* share 53% identity. *smORF72* and *smORF63* share only 11% identity. As mentioned in the response to the Reviewer's second major comment, we used AlphaFold to predict the *TF72* and *TF63* structures and to probe for structural similarities between them and other proteins. Additionally, AlphaFold modeling predicted the residues in each TF that comprise the DNA-recognition helix. Informed by this prediction, we exchanged 5 residues between *TF72* and *TF63*. These residues are located within the recognition helix and the flanking C-terminal loop in each TF. Remarkably, exchange of only those 5 residues between the two TFs reversed their preferences for *smORF72* and *smORF63* promoter DNA (Figure 2b). The text describing the work is in lines 204-207. We believe these newly added experiments deliver key information regarding the molecular basis for the exquisite specificity of the two phage pathways.

We additionally note that the most significant divergence between *TF63* and *TF72* occurs in alpha helices 6 and 7. These helices are clearly outside the DNA-binding domain and distinguish these phage TFs from the other near-neighbor structural matches. We speculate that these helices are important for cue detection/ligand binding. These most highly divergent regions could underpin cue differentiation by phage 63 and phage 72.

We thank this Reviewer for inspiring us to delve more deeply into the topic as it has improved the quality of the manuscript and led to new hypotheses for us to examine in the future.

5) line 152-162 - here the authors carry out a simple lux reporter assay to track expression of the *pR72* promoter, and show that *cl72* expression decreases and *smORF72* increases expression levels from the *pR72* promoter. The authors need to phrase their findings more carefully given there is no biochemical data provided to demonstrate e.g. that *cl72* acts on the *pR72* promoter

(line 155), or that smORF72 acts directly to inactivate cl72 (line 162). None of this is directly demonstrated, and therefore I feel the authors are over-interpreting the data.

As requested, we purified the cl₇₂ protein and showed that it shifts P_{R72} promoter DNA using an EMSA. The experiment is described in lines 139-140 and the data are shown in Extended Data Figure 1g. We also performed the analogous experiment for each of the other repressors and their target promoters under study (purified cl₆₃ protein and P_{R63} promoter DNA, purified cl_{Apop} protein and P_{R-Apop} promoter DNA, and purified cl_{ARM81ld} protein and P_{R-ARM81ld} promoter DNA). These gels now appear as Extended Data Figures 4f, 6d, 7c, respectively.

We have previously shown that Qtip binds to and inactivates cl_{VP882} using biochemical approaches. Our new data concerning smORF₇₂, smORF₆₃, cl₇₂, and cl₆₃ suggest an analogous mechanism. However, to date both the smORF₇₂ and smORF₆₃ proteins remain highly intractable to purification and analysis. Thus, their characterization must await future work. We take the Reviewer's caution and we have softened our conclusions (lines 142-144).

6) line 164-172 the authors use a lux reporter assay to show that TF72 expression increases smORF72 expression (28-fold) in *E. coli* and in 1F-97 (7-fold), where it induces lysis. Again, no biochemistry, so bit more care in the way things are phrased is recommended.

We successfully expressed, purified, and demonstrated in vitro DNA-binding of the cl repressors to address the Reviewer's above comment (point 5), however, we were unable, despite numerous attempts to sufficiently overexpress the TFs to enable purification.

For these reasons, we toned down the statement that TF₇₂ is the direct activator of smORF₇₂ expression as the Reviewer suggested. We note that in the revised submission, we have included new reporter data that reveals the promoter specificities of the two *Vibrio* 1F-97 TFs and that specificity can be reversed by exchange of 5 residues in the predicted DNA-binding domains. We hope these new efforts, the new results, and our explanations can satisfy the Reviewer on this point.

7) No information is given about the similarity (nucleotide, amino acid, predicted Alphafold structures) of the TF₆₃/TF₇₂ and smORF₆₃/smORF₇₂ genes. From the text, I have the impression they are extremely closely related. Can the authors provide some alignments/structure predictions to clarify this?

In the revised work, we now provide additional figures with amino acid sequence alignments for TF₇₂ and TF₆₃ and for smORF₇₂ and smORF₆₃ (Extended Data Figure 3a and 3e, respectively), as well as AlphaFold predictions for TF₇₂ and TF₆₃ (Extended Data Figure 5a and 5b).

8) Throughout, the authors focus all their assays on prophage induction, and never mention or measure lysogenization. Why is this? Is there a reason to assume the regulators are / aren't involved in this?

The Reviewer is correct, and this is a main goal of the lab. We have not yet successfully developed assays to study infection by these phages. There is no reason to assume that the regulators do

not play key roles during steps in infection beyond induction of lysis. We make these points in the revised Discussion (lines 395-397 and 430-432).

Referee #2 (Remarks to the Author):

Major Comments

A combination of bioinformatics and molecular genetics are used to identify a transcription factor and small protein circuit controlling the lysis decision in multiple prophages. This circuit is compared to the classical SOS response induction pathway (i.e., RecA). This work is an extension of the quorum sensing (QS)-induction work and illuminates several key components.

The paper is predicated on the premise that the phage field ubiquitously believes that DNA damage is the primary induction signal. This is an old view and the phage ecology literature is replete with models and experiments showing that DNA damage is not the major inducing factor. The manuscript would have more impact if the emphasis was placed on the "phage-on-phage war".

The Reviewer makes excellent points. Regarding DNA damage, we agree that, increasingly there are examples of non-SOS pathways to induction. We now explain that RecA-independent phage induction has been shown to occur in lambdoid phages in *E. coli*, however the molecular mechanisms underlying these alternative pathways are unknown. That and a citation are in lines 378-379 of the Discussion.

Regarding the Reviewer's comment concerning phage-on-phage war, we are very grateful to be pointed in that direction. The Reviewer inspired us to substantially alter the emphasis of the manuscript, from characterization of novel TF-smORF modules, to inter-prophage competition. This new slant provides superior context for why we searched for SOS-independent regulators of phage lysis. Additionally, we have expanded the conceptual novelty of the work by experimentally exploring phage-on-phage "war" by including RNA FISH experiments to visualize, for the first time, co-induction of phage 72 and phage 63 following DNA damage or induction via smORF₇₂ or smORF₆₃ overproduction. We also performed and provided analogous experiments for the *Aeromonas* sp. ARM81 phages. Interestingly, in the case of DNA-damage, in which both phages respond, we find that discrete subsets of cells produce either one phage, the other phage, or both phages. Thus, single and dual phage production is possible from an individual host cell. The new RNA FISH experiments are included in lines 321-345 of the manuscript and the data are shown in Figure 4b and 4c.

Overall, this is an intriguing work that is potentially of broad interest.

All of the "we" are distracting. Right now, the paper reads like an undergraduate discussing a night out... "We wondered", "We could not", "We synthesized", "We..." makes it almost impossible to follow the narrative. Similarly, the figures need to be improved. Add legends to the growth

curves. I am a knowledgeable reader (i.e., I know main proteins & background for this work) and I had a very hard time understanding what you did.

As requested, we edited the manuscript to remove many of the instances of “We,” and we have added keys to all growth and lysis curves.

I really can't tell whether the data points are true replicates (i.e., from individual colonies on different days) or pseudoreplicates (i.e., the same culture sampled 3 times)? I think it is the latter, which means everything was effectively done from 1 colony to one overnight...Also, the experiments need to be repeated at least 5 times to make them more statistically sound.

Indeed, the data points are true replicates from individual colonies. In the original manuscript and in the revised version, we noted in all figure legends that data were collected from $n = 3$ biological replicates. To clarify, we have now written in the Methods section under “Bacterial strains and growth conditions” that all experiments were conducted in triplicate from three independent colonies.

Minor Comments

How common is the small protein (smORF) in the Short Read Archive?

We elected not to conduct this search with *qtip* or *smORF*₇₂, based on the near-complete absence of homologs in the NCBI NR database (reported by our group for *qtip* in 2019). A similar query with *smORF*₆₃ did reveal a few homologs in the NR database but these hits were entirely within *Vibrionaceae*. Perhaps, unsurprisingly, they also overlapped significantly with the hits generated from a TF₆₃-based query performed in response to a comment by Reviewer 3. These findings are now reported in Supplementary Table 2. For these reasons, we anticipate that an SRA query would yield limited new information, as the sequencing projects with substantial smORF-mappable reads are likely to be those replete with *Vibrionaceae*.

There are several papers exploring novel inducing agents. It would be interesting to see which pathway these other agents cause induction.

We agree with the Reviewer that knowledge of the cues to which the TF-smORFs studied here respond is of utmost importance. Indeed, identifying the native cues controlling expression and activity of these modules is a primary focus of our ongoing research, but thus far, our efforts have been unsuccessful. We have screened commercially available compound libraries (Biolog MicroArrays, ~2,200 conditions) and a curated library of antibiotics (Seyedsayamdost Group, ~250 conditions) but we have not yet identified an inducer. In the revised manuscript, we now include the negative data and the details of the screens in Extended Data Figures 2a and 4g.

Referee #3 (Remarks to the Author):

Summary of the article:

These authors discover and characterize novel lysis regulatory modules in linear plasmid-like phages. Inspired by the arrangement of a previously discovered regulatory module (VqmAphage and qtip), they bioinformatically searched through sequence databases and discovered several instances of a divergently transcribed transcription factor and a small open reading frame (smORF). Through heterologous expression of the genes singly and in combination in *E. coli*, they find that the smORF non-proteolytically interferes with the *cl* repressor of the phage that carries it, leading to phage induction. Through confocal microscopy, they show that the two smORFs from the two phages in the *Vibrio* strain they use act on the *c*-repressors in different ways. Further bioinformatic searches reveal TF-smORF modules that include a *luxR* gene, which they show is responsive to host-produced quorum-sensing molecules. Finally, they show that in polylysogens which contain these smORF-activated modules, expression of the smORF on one phage leads to near-exclusive production of that virus, creating a fitness advantage for that virus in that inducing condition. These results will be of interest to a more specific audience of those interested in the mechanisms of phage-host relationships and their implied ecological consequences.

The manuscript is clear and enjoyable (although long) read. The most exciting part of the paper is that it demonstrates a novel, phage specific, DNA-damage-independent mode of viral induction from bacterial genomes. The data to support this finding are compelling. Unfortunately, the clarity, elegance and impact of the story seem to be lost in the writing that combines two to three papers into one long and circuitous one.

The title, introduction, and discussion fit with the story that is very nicely told through the paragraph starting line 233. Then a long detour begins where the authors introduce a totally new concept about the sequestering of the repressor to the cell poles and use Halo-tag to identify the localization of smORF72. This detour seems out of place and continues into a third direction with the XRE-LuxR genes of other phage in other species. As someone who is not an expert in the genetic methods, I felt these detours are unnecessary and story really loses momentum. The specificity and phage induction in a polylysogen which is FINALLY stated in the paragraph starting 451. While I was reading it I felt such relief when the authors came back around to their message. I had almost given up. I suggest cutting out the detours and sticking to one story for those who haven't read all of the other work.

We agree with the Reviewer that the primary goal of our work, to study polylysogeny and inter-phage competition, was not clearly communicated in the initial manuscript. In response, we have substantially altered the emphasis of the manuscript, from characterization of novel TF-smORF modules to inter-phage competition. This new slant provides superior context for why we searched for SOS-independent regulators of phage lysis. Additionally, we have expanded the conceptual novelty of the work by experimentally exploring phage-on-phage "war" by including RNA FISH experiments to visualize, for the first time, co-induction of phage 72 and phage 63 following DNA damage or induction via smORF₇₂ or smORF₆₃ overproduction. We also performed and provided analogous experiments for the *Aeromonas* sp. ARM81 phages. Interestingly, in the

case of DNA-damage, in which both phages respond, we find that discrete subsets of cells produce either one phage, the other phage, or both phages. Thus, single and dual phage production is possible from an individual host cell. The new RNA FISH experiments are included in lines 321-345 of the manuscript and the data are shown in Figure 4b and 4c.

Other comments:

- I found the logic stated on lines 75 – 78 for focusing on phage with RecA-dependent, autoproteolytic *cl* repressors is not clear. I read this several times and do not follow this reasoning for excluding other phage. Isn't diversity being lost?

We agree with the Reviewer that our explanation for focusing on autoproteolytic *cl* repressors should be clearer. We have expanded the text and clarified our logic (lines 87-94).

We also acknowledge that diversity is sacrificed by the constraints of our search (lines 375-377 of the revised Discussion).

- In a related question, on line 86: the statement “indicating that our autoproteolytic set less redundantly sampled nature’s diversity” is unclear. Do you mean that less than what? The authors must know that the databases do not sample viruses or microbes in an unbiased way. This tells us nothing about “Nature’s diversity” and the concept of “less redundantly” is very confusing. Please restate.

The Reviewer makes a good point. In response to this comment and Reviewer 1’s suggestion to streamline this section, we have removed these sentences from the manuscript.

- The frequency of other small-ORF candidates throughout phage genomes, independently of gene neighborhood, was not assessed. The logic for keeping the search limited to gene neighborhoods is not clear. If the average length of these phages is estimated at around 50kb (and at least <100kb), casting a 10kb net around a *parB* sequence encompasses 40% of that prophage prediction, which decreases the chance that *parB* presence/absence is coupled to these regulatory modules. Line 253 cites the use of 25kb as a cutoff for searching for modules near “conserved phage structural genes”. Many *c*-repressors are more than within that cutoff, making its use a confusing filter. How many small ORF/transcription pairs can be found without requiring an association with *parB* or *repA*, or when the presence of a different gene (like an integrase) is required? In general, the discovery of a TF-smORF pair that was not coupled with the originally queried *repA*-*telN* proteins lends itself to a broader search, not an additional, equally limiting search. These filtration steps are used as a lead-in to the discovery of the already-discovered and ready-to-use *Aeromonads*, which seems convenient and diminishes the integrity of the posed question. Additionally, the emphasis on “small protein modules” as suggested by the title is not reflected in the way these searches were conducted.

To address this issue, we now include a detailed description of the TF₆₃-based query we carried out to discover lysis-controlling modules in phage genomes, irrespective of genomic context. We also fully describe our logic for progressing from these preliminary BLAST-based searches to the *ParB*-based “guilt-by-association” search strategy. We appreciate this comment as we now

realize that, in the service of brevity, omission of the logic progression in the initial manuscript was confusing. The *Aeromonas* discoveries do arise from these queries, but the searches were not orchestrated to achieve this result. The hits from the preliminary TF₆₃ search are provided in Supplementary Table 2 and the final ParB-associated loci described in Supplementary Tables 1 and 3.

- There is no discussion of the integrity of the sequences that are put forward as putative phage genomes, at least in the NCBI nt database. The additional step at least of phage prediction from sequences of interest could have cut down on some of the need for arbitrary distance cutoffs. The lack of reference to recognizable lysogeny genes (integrases, transposases, c-repressors) in a paper about putative induction modules is glaring, as searches for putative induction machinery implies successful latent infections must have taken place. This problem is implicitly overcome by synthetic generation of gene cassettes in some cases, but should be explicitly referenced.

We thank the Reviewer for this idea. We have now used the phage prediction tool VIBRANT to confirm that all reported sequences from our RepA-TelN, ParB, and TF₆₃-based queries are likely to originate from phage genomes. We report this analysis in the main text. The relevant information from the analysis is provided in Supplementary Table 1 and in our updated Methods section.

- Line 374: The heading “Variations in host bacterial signal transduction cascades...” implies that some phages have a “choice” of what transcription factor to use, where different transcription factors encoded by the same phage may act independently on these regulatory modules. However, the bioinformatic results of the XRE_{Apop} tBLASTn search were not clearly described. Were only five viruses subsequently found, as described in the Extended Data Fig 3a? The question posed on line 377 has already been disproven earlier in the paper, which show extensive phage TF-smORF modules without a LuxR component. The XRE-LuxR modules were happened upon ostensibly as one of many TF-smORF pairs, and their treatment as a unique category limits the potential directions of the sensory inputs these phages are using. Any cursory investigation into *Shigella* quorum-sensing pathways which would reveal homologues on *Shigella* phages would be a more satisfying reason to include this section of the paper. Without it, the conclusion on line 395 is mystifyingly obvious.

The Reviewer makes several excellent points. Given our goal of streamlining the characterization of TF-smORFs and the new focus on inter-prophage competition in polylysogens, we have removed this section from the Results. We now only briefly mention the phage-encoded XREs in *Shigella* genomes in the Discussion in lines 420-426.

The goal of our tBLASTn search was to determine if XRE and LuxR phage components always function hand-in-hand, analogous to how they function in the *Aeromonas* phages studied here. Our finding that phage-encoded XRE proteins are present in *Shigella* phages, in the absence of partner LuxR proteins, indicates that XRE and LuxR are not an obligate pair in phage TF-smORF modules. To address the Reviewer’s questions directly, the three *Shigella* XREs were the closest entries, outside of *Aeromonas*, to XRE_{Apop}. Inspection of the genomic contexts of these contigs led us to hypothesize that all three are linear plasmid-like *Shigella* phages. That finding motivated us to test the relevant regulatory principles by gene/promoter synthesis. There were higher-scoring entries to XRE_{Apop} within other *Aeromonas* species and lower-scoring entries outside of

Aeromonas and *Shigella*. We did not pursue these genes further as XRE-type domains are a broad class of DNA-binding proteins and the entries lacked strong evidence of being phage-encoded. Again, to re-focus the manuscript on inter-prophage competition, we have removed this section from the Results and included minimal text in the Discussion to note that XRE and LuxR phage proteins do not always co-occur.

- Given that this work was done from a pool of “61 unique loci” and “56 unique contigs” and “three contigs, likely from linear plasmid-like phages” it seems unlikely that at this stage it is reasonable to say on line 520 that “these pathways may in fact be the primary determinants of phage lifestyle transitions”. This work seems to specifically interrogate linear plasmid-like phages and doesn’t attempt to assess phages writ broad. What is known about the frequency and abundance of this phage type in nature? A frank discussion about this would be useful given the specificity of the findings. Line 377 admits that the investigation has taken place in “particular sets of regulatory components encoded on different plasmid-like phages”, limiting the applicability of this work to phages in general.

As requested, we have softened our claim on lines 409-412.

We agree with the Reviewer that our work focuses on a rarer subset of phages. However, phages, in general, often co-exist as polylysogens and compete for host resources. While our mechanistic understanding of phage-specific induction modules is limited to plasmid-like phages, the general principles of phage-on-phage competition we uncovered from studying the *Vibrio* 1F-97 and *Aeromonas* sp. ARM81 lysogens could apply more broadly. Moreover, the development of RNA FISH analyses to visualize prophage induction could be applied to the investigation of many phages during infection or induction. In addition, and as noted above, we now include sentences in the Discussion to explain our rationale for the search (lines 375-377).

- The final paragraph of the discussion is an incompletely described “just so story” about the relevance of this mechanism to the phage. This is at best speculative and should absolutely be removed as it only hurts the impact of the paper overall.

We have removed these sentences as requested.

Minor comments:

- Figure 2a could have been easily used as well to illustrate the lack of identity using Mauve or Clinker (helpful tool – see Gilchrist and Chooi, 2020, Bioinformatics). How similar are repA, telN, and the partition machinery?

The genes of interest on phage 72 and phage 63 were identified based on synteny combined with our genetic and biochemical experiments. All analogous genes on the two phages share very low homology (<30% amino acid identity) and thus fail to align using algorithms such as Clinker. The only exception are the TFs. They share 54% amino acid identity and are indeed homologs. We have provided the statistics below in answer to the Reviewer’s question on gene-by-gene identity, and we have introduced a new line in the manuscript and in the Figure 2a legend summarizing this point.

Reciprocal BLASTp results:

query: subject: phage 72: phage 63

telN: 29.8% ID over 26% query

tf: 54.3% ID over 94% query

smORF: No significant identify detected

repA: 20.7% ID over 56% query

cl: 25.6%ID over 90% query

parA: 25.2% ID over 44% query

parB: No significant identify detected

query: subject: phage 63: phage 72

telN:27.0% ID over 37% query

tf: 54.3% ID over 96% query

repA: 20.7% ID over 56% query

cl: 25.6% ID over 94% query

parA: 25.2% ID over 36% query

- Line 189: share little identity..(Mauve), is this supposed to be a figure? A reference? A figure would be nice.

The lack of identity between the phages across their genomes, with the exception of the *tf* genes, makes Mauve or Clinker unhelpful in providing additional insight on genome-wide similarities. We have provided the gene-by-gene identity statistics in the above response. To the manuscript, we have added the summary statistics, and we note the lack of identity between the phages in the main text and in the caption to Figure 2a.

- Line 410: Figure 2a is referenced to support amino acid identity, but does not contain any amino acid identity information.

Yes, a good point. We have now provided additional figures with amino acid sequence alignments for TF₇₂ and TF₆₃ and smORF₇₂ and smORF₆₃ (Extended Data Figure 3a and 3e, respectively).

- The presence of autoproteolytic repressors is an interesting filter to use, but was not repeated in the search for parB-associated loci and could have been useful.

The primary motivation to use an autoproteolytic filter in our RepA-TelN dataset was to eliminate the preponderance of *Enterobacteriaceae* N15-like sequences from this set that were unlikely to be of further interest to our investigation (N15-like phages carry non-autoproteolytic repressors).

This phenomenon was less problematic in the ParB-associated dataset and so we decided against including this additional data filter.

- The strain that was previously studied (VP882) is included in the tree in Fig 1A, but the *Vibrio cyclitrophicus* 1F-97 strain that is a major subject of study in this paper does not seem to be included in the tree. By contrast, the other investigated phages are included in the tree when they're referenced.

We apologize for this omission and thank the Reviewer for highlighting our oversight. The 1F-97 strain has always been present in this figure (accession KP795532), and is now explicitly indicated with a star.

- Lines 511-519 (“Presumably, the native cues... niche in which they reside”) are confusing in the context of phage induction, as the first cyanophage example suggests induction would not be favored, and the second suggests a purely lytic relationship with a bacterial host, without any need for multiple, finely-tuned induction pathways. Neither of them are suggestive of the dynamic relationship that is proposed to govern these phages, nor do they suggest what the referenced cues might be. See above.

As noted above, we have removed these sentences from the Discussion.

- Line 3 is incorrect in light of several recent works highlighting the importance of chronic viruses in several different taxa, and is not a strong beginning to a strong paper. See especially, among several others, Roux et al, 2019 in Nature Microbiology.

The Reviewer makes a good point concerning the importance of chronic viruses. We have modified the first paragraph of the Introduction to note that phages can cause chronic infections in which they persist and extrude viral particles without killing the host.

Reviewer Reports on the First Revision:

Referees' comments:

Referee #1 (Remarks to the Author):

I am satisfied by the way my comments have been addressed.

Referee #3 (Remarks to the Author):

This was a solid article that is now improved in revision. All of our comments have been addressed.

Author Rebuttals to First Revision:

Referee #1 (Remarks to the Author):

I am satisfied by the way my comments have been addressed.

Thank you. We are happy that we were able to address your concerns.

Referee #2 (noted as "Comments from Referee #2" in your decision letter):

The reviewer indicated that they remained positive about the manuscript, but found the statistics troubling. In particular, this reviewer said it would be important to clarify the following points in the main manuscript:

We are pleased that the reviewer was positive about the work. Below, we clarify the remaining issues concerning statistics.

* In the rebuttal letter, the author's state that there were 3 biological replicates, which is good, but they did not add this information to the article as indicated in the response to reviewers. Please confirm the number of replicates for all experiments in the figure legends and the Methods.

The numbers of biological replicates for each panel are indeed provided in the final line of the figure legend. In the revised Methods, we also now state the number of replicates in the section "Quantitation and statistical analyses". We have also made the source data for all points in each panel available and those data provide the results from each replicate.

* It is also unclear whether the tests were conducted on the same day or whether they were completely independent runs. Please clarify this in the main manuscript.

As is now noted in the Methods, growth, lysis, and reporter assays were conducted on the same day starting from overnight cultures derived from three biologically independent bacterial colonies. This method is standard practice in the field and used by our and, indeed many other, labs. For example, see: Jiao, C., Reckstadt, C., König, F. et al. *Nat Biotechnol* (2023), <https://www.nature.com/articles/s41587-022-01604-8>. Following the *Nature* guidelines for figure legends, and after verifying what text goes where with the Editor, the number of biological replicates is reported in the final line of each legend. Experimental details are described in the Methods (in the newly added section "Quantitation and statistical analyses", as noted in the above response). We hope this clarifies the point.

Referee #3 (Remarks to the Author):

This was a solid article that is now improved in revision. All of our comments have been addressed.

Thank you for your thoughtful feedback The additional experiments we performed in response to your comments expanded our own understanding of our system, which should be appreciated by readers.